# Flipping-based Policy for Chance-Constrained Markov Decision Processes

**Xun Shen**
Osaka University
shenxun@eei.eng.osaka-u.ac.jp

**Shuo Jiang**
Osaka University
u316354h@ecs.osaka-u.ac.jp

**Akifumi Wachi**
LY Corporation
akifumi.wachi@lycorp.co.jp

**Kazumune Hashimoto**
Osaka University
hashimoto@eei.eng.osaka-u.ac.jp

**Sebastien Gros**
Norwegian University of Science and Technology
sebastien.gros@ntnu.no

## Abstract

Safe reinforcement learning (RL) is a promising approach for many real-world decision-making problems where ensuring safety is a critical necessity. In safe RL research, while expected cumulative safety constraints (ECSCs) are typically the first choices, chance constraints are often more pragmatic for incorporating safety under uncertainties. This paper proposes a *flipping-based policy* for Chance-Constrained Markov Decision Processes (CCMDPs). The flipping-based policy selects the next action by tossing a potentially distorted coin between two action candidates. The probability of the flip and the two action candidates vary depending on the state. We establish a Bellman equation for CCMDPs and further prove the existence of a flipping-based policy within the optimal solution sets. Since solving the problem with joint chance constraints is challenging in practice, we then prove that joint chance constraints can be approximated into Expected Cumulative Safety Constraints (ECSCs) and that there exists a flipping-based policy in the optimal solution sets for constrained MDPs with ECSCs. As a specific instance of practical implementations, we present a framework for adapting constrained policy optimization to train a flipping-based policy. This framework can be applied to other safe RL algorithms. We demonstrate that the flipping-based policy can improve the performance of the existing safe RL algorithms under the same limits of safety constraints on Safety Gym benchmarks.

## 1 Introduction

In safety-critical decision-making problems, such as healthcare, economics, and autonomous driving, it is fundamentally necessary to consider safety requirements in the operation of physical systems to avoid posing risks to humans or other objects [14, 19, 43]. Thus, safe reinforcement learning (RL), which incorporates safety in learning problems [19], has recently received significant attention for ensuring the safety of learned policies during the operation phases. Safe RL is typically addressed by formulating a constrained RL problem in which the policy is optimized subject to safety constraints [1, 15, 32, 49]. The safety constraints have various types of representations (e.g., expected cumulative safety constraint [4, 5, 7], instantaneous hard constraint [36, 45], almost surely safe [9, 44], joint chance constraint [29–31]). In many real applications, such as drone trajectory planning [40] and

38th Conference on Neural Information Processing Systems (NeurIPS 2024).

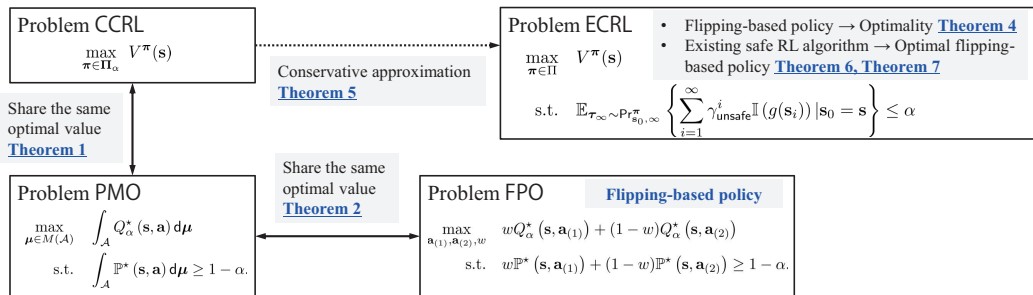

Figure 1: Summary of the relations among main theorems and problems in this paper.

planetary exploration [8], safety requirements must be satisfied at least with high probability for a finite time mission, where joint chance constraint is the desirable representation [33, 43].

**Related work.** The optimal policy for RL without constraints or with hard constraints is deterministic policy [10, 20, 39]. Introducing stochasticity into the policy can facilitate exploration [11, 38] and fundamentally alter the optimization process during training [16, 21, 37], affecting how policy gradients are computed and how the agent learns to make decisions. It has been shown that the optimal policy for a Markov decision process with expected cumulative safety constraints is always stochastic when the state and action spaces are countable [3]. Policy-splitting method has been proposed to optimize the stochastic policy for safe RL with finite state and action spaces [12]. In [35], an algorithm was proposed to compute a stochastic policy that outperforms a deterministic policy under chance constraints, given a known dynamical model. In more general settings for safe reinforcement learning, such as with uncountable state and action spaces, the theoretical foundation regarding whether and how a stochastic policy can outperform a deterministic policy under chance constraints remains an open problem. Developing practical algorithms to obtain optimal stochastic policies with chance constraints requires further investigation.

**Contributions.** We present a Bellman equation for CCMDPs and prove that a flipping-based policy archives the optimality for CCMDPs. Flipping-based policy selects the next action by tossing a potentially distorted coin between two action candidates where the flip probability and the two candidates depend on the state. While solving the problem with joint chance constraints is computationally challenging, the problem with the Expected Cumulative Safe Constraints (ECSCs) can be effectively solved by many existing safe RL algorithms, such as Constrained Policy Optimization (CPO, [1]). Thus, we establish a theory of conservatively approximating the joint chance constraints by ECSCs. We further show that a flipping-based policy achieves optimality for MDP with ECSCs. Leveraging the existing safe RL algorithms to obtain a conservative approximation of the optimal flipping-based policy with chance constraints is possible. Specifically, we present a framework for adapting CPO to train a flipping-based policy using existing safe RL algorithms. Finally, we show that our proposed flipping-based policy can improve the performance of the existing safe RL algorithms under the same limits of safety constraints on Safety Gym benchmarks. Figure 1 summarizes the main contributions.

## 2 Preliminaries: Markov Decision Process

A standard Markov decision process (MDP) is defined as a tuple, $\langle \mathcal{S}, \mathcal{A}, r, \mathcal{T}, \mu_0 \rangle$. Here, $\mathcal{S}$ is the set of states, $\mathcal{A}$ is the set of actions, $r : \mathcal{S} \times \mathcal{A} \to \mathbb{R}$ is the reward function. This paper considers the general case with state and action sets in finite-dimension Euclidean space, which can be continuous or discrete. Let $\mathcal{B}(\cdot)$ be the Borel $\sigma$-algebra on a metric space and $M(\cdot)$ be the set of all probability measures defined on the corresponding Borel space. The state transition model $\mathcal{T} : \mathcal{S} \times \mathcal{A} \to M(\mathcal{S})$ specifies a probability measure of a successor state $\mathbf{s}^+$ defined on $\mathcal{B}(\mathcal{S})$ conditioned on a pair of state and action, $(\mathbf{s}, \mathbf{a}) \in \mathcal{S} \times \mathcal{A}$, at the previous step. Specifically, we use $p(\cdot|\mathbf{s}, \mathbf{a})$ to define a conditional probability density associated with the state transition model $\mathcal{T}(\mathbf{s}, \mathbf{a})$. Finally, $\mu_0$ is the distribution of the initial state $\mathbf{s}_0 \in \mathcal{S}$. A stationary policy $\kappa : \mathcal{S} \to M(\mathcal{A})$ is a map from states to probability measures on $(\mathcal{A}, \mathcal{B}(\mathcal{A}))$. We use $\boldsymbol{\pi}(\cdot|\mathbf{s})$ to define a conditional probability density associated with $\kappa(\mathbf{s})$, which specifies the *stationary policy*. Define a trajectory in the infinite horizon by $\boldsymbol{\tau}_\infty := \{\mathbf{s}_0, \mathbf{a}_0, \mathbf{s}_1, \mathbf{a}_1, ..., \mathbf{s}_k, \mathbf{a}_k, ...\}$. An initial state $\mathbf{s}_0$ and a stationary policy $\boldsymbol{\pi}$ defines a unique probability measure $\mathrm{Pr}^{\boldsymbol{\pi}}_{\mathbf{s}_0, \infty}$ on the set $(\mathcal{S} \times \mathcal{A})^\infty$ of the trajectory $\boldsymbol{\tau}_\infty$

[22]. The expectation associated with $\text{Pr}^{\boldsymbol{\pi}}_{\mathbf{s}_0,\infty}$ is defined as $\mathbb{E}_{\boldsymbol{\tau}_\infty \sim \text{Pr}^{\boldsymbol{\pi}}_{\mathbf{s}_0,\infty}}$. Given a policy $\boldsymbol{\pi} \in \Pi$, the value function at an initial state $\mathbf{s}_0 = \mathbf{s}$ is defined by $V^{\boldsymbol{\pi}}(\mathbf{s}) := \mathbb{E}_{\boldsymbol{\tau}_\infty \sim \text{Pr}^{\boldsymbol{\pi}}_{\mathbf{s}_0,\infty}} \{R(\boldsymbol{\tau}_\infty) \mid \mathbf{s}_0 = \mathbf{s}\}$ with $R(\boldsymbol{\tau}_\infty) := \sum_{k=0}^{\infty} \gamma^k r(\mathbf{s}_k, \mathbf{a}_k)$, where $\gamma \in (0,1)$ as the discount factor. Also, the action-value function is defined as $Q^{\boldsymbol{\pi}}(\mathbf{s}, \mathbf{a}) := r(\mathbf{s}, \mathbf{a}) + \gamma \mathbb{E}_{\boldsymbol{\tau}_\infty \sim \text{Pr}^{\boldsymbol{\pi}}_{\mathbf{s}_0,\infty}} \{V^{\boldsymbol{\pi}}(\mathbf{s}^+) \mid \mathbf{s}_0 = \mathbf{s}, \mathbf{a}_0 = \mathbf{a}\}$.

## 3 Flipping-based Policy with Chance Constraints

A constrained Markov decision process (CMDP) is an MDP equipped with constraints restricting the set of policies. Let $\mathbb{S}$ be the "safe" region of the state specified by a continuous function $g : \mathcal{S} \to \mathbb{R}$ in the following way: $\mathbb{S} := \{\mathbf{s} \in \mathcal{S} : g(\mathbf{s}) \le 0\}$. Let $T \in \mathbb{N}_+$ be the episode length. As suggested in [30, 31], the following joint chance constraints is imposed:

$$\text{Pr}^{\boldsymbol{\pi}}_{\mathbf{s}_0,\infty} \{\mathbf{s}_{k+i} \in \mathbb{S}, \forall i \in [T] \mid \mathbf{s}_k \in \mathbb{S}\} \ge 1 - \alpha, \ \forall k = 0, 1, 2, ... \tag{1}$$

where $\alpha \in [0,1)$ denotes a safety threshold regarding the probability of the agent going to an unsafe region and $[T] := \{1, ..., T\}$ is the index set. The left side of the chance constraint (1) is a conditional probability, specifying the probability of having states of future $T$ steps in the safe region when $\mathbf{s}_k$ is inside the safe region. When the system is involved with unbounded uncertainty $\mathbf{w}$, it is impossible to ensure the safety with a given probability level in infinite-time scale [18]. Instead, ensuring safety in a future finite time when the current state is within the safety region is reasonable and practical [20]. This paper calls the MDP equipped with chance constraint (1) as Chance Constrained Markov decision processes (CCMDPs). It refers to the problem with almost surely safe constraint when $\alpha = 0$. The set of feasible stationary policies for a CCMDP is defined by $\boldsymbol{\Pi}_\alpha := \{\boldsymbol{\pi} \in \boldsymbol{\Pi} : \forall \mathbf{s}_k \in \mathbb{S}, (1) \text{ holds}\}$. Chance constrained reinforcement learning (CCRL) for a CCMDP is to seek an optimal constrained stationary policy by solving

$$\max_{\boldsymbol{\pi} \in \boldsymbol{\Pi}_\alpha} V^{\boldsymbol{\pi}}(\mathbf{s}). \tag{CCRL}$$

Define optimal solution set of Problem CCRL by $\Pi^\star_\alpha := \{\boldsymbol{\pi} \in \Pi_\alpha : V^{\boldsymbol{\pi}}(\mathbf{s}) = \max_{\boldsymbol{\pi} \in \Pi_\alpha} V^{\boldsymbol{\pi}}(\mathbf{s})\}$. Let $\boldsymbol{\pi}^\star_\alpha \in \Pi^\star_\alpha$ be an optimal solution of Problem CCRL. Associated with $\boldsymbol{\pi}^\star_\alpha$, we denote $V^\star_\alpha(\mathbf{s}) := V^{\boldsymbol{\pi}^\star_\alpha}(\mathbf{s})$ and $Q^\star_\alpha(\mathbf{s}, \mathbf{a}) := Q^{\boldsymbol{\pi}^\star_\alpha}(\mathbf{s}, \mathbf{a})$ for the optimal value and value-action functions.

Define a function $\mathbb{P}^\star(\mathbf{s}, \mathbf{a}) := \text{Pr}^{\boldsymbol{\pi}^\star_\alpha}_{\mathbf{s},\infty} \{\mathbf{s}_{k+i} \in \mathbb{S}, \forall i \in [T] \mid \mathbf{s}_k = \mathbf{s}, \mathbf{a}_k = \mathbf{a}\}$. The continuity of $\mathbb{P}^\star(\mathbf{s}, \mathbf{a})$ is guaranteed under mild conditions giving as Assumptions 1 and 2 (pp. 78-79 of [25]). Besides, the upper semicontinuity of $Q^\star_\alpha(\mathbf{s}, \mathbf{a})$ is from Assumption 1.

**Assumption 1.** *Suppose that $\mathcal{A}$ is compact and $r(\mathbf{s}, \mathbf{a})$ is continuous[1] on $\mathcal{S} \times \mathcal{A}$. Besides, assume that the state transition model $\mathcal{T}$ can be equivalently described by $\mathbf{s}^+ = f(\mathbf{s}, \mathbf{a}, \mathbf{w})$, where $\mathbf{w} \in \mathcal{W} \subseteq \mathbb{R}^s$ is a random variable and $f(\cdot)$ is a continuous function on $\mathcal{S} \times \mathcal{A} \times \mathcal{W}$. The probability density function is $p_{\mathbf{W}}(\mathbf{w})$.*

Assumption 1 is natural since it only requires that the reward function is continuous and the state transition can be specified by a state space model with a continuous state equation, which is general in many applications. We do not require $f(\cdot)$ to be available.

**Assumption 2.** *The constraint function $g(\cdot)$ is continuous. For every $\mathbf{s} \in \mathcal{S}$ and $\mathbf{a} \in \mathcal{A}$, we have*

$$\text{Pr}^{\boldsymbol{\pi}^\star_\alpha}_{\mathbf{s},\infty} \left\{\max_{i \in [T]} g(\mathbf{s}_{k+i}) = 0 \mid \mathbf{s}_k = \mathbf{s}, \mathbf{a}_k = \mathbf{a}\right\} = 0. \tag{2}$$

Assumptions 1 and 2 are essentially assuming the regularities of $g$ and $\mathcal{T}(\mathbf{s}, \mathbf{a})$, which is not a strong assumption. With $\mathbb{P}^\star(\mathbf{s}, \mathbf{a})$, we define a probability measure optimization problem (PMO) by

$$\max_{\boldsymbol{\mu} \in M(\mathcal{A})} \int_{\mathcal{A}} Q^\star_\alpha(\mathbf{s}, \mathbf{a}) \, \mathsf{d}\boldsymbol{\mu} \quad \text{s.t.} \quad \int_{\mathcal{A}} \mathbb{P}^\star(\mathbf{s}, \mathbf{a}) \, \mathsf{d}\boldsymbol{\mu} \ge 1 - \alpha. \tag{PMO}$$

We have the following theorem for the optimal constrained stationary policy $\boldsymbol{\pi}^\star_\alpha$ of Problem CCRL:

**Theorem 1.** *The optimal value of Problem PMO equals $V^\star_\alpha(\mathbf{s})$ for any $\mathbf{s} \in \mathcal{S}$. The probability measure $\boldsymbol{\mu}^\star_\alpha$ associated with $\boldsymbol{\pi}^\star_\alpha(\cdot|\mathbf{s})$ is an optimal solution of Problem PMO for any $\mathbf{s} \in \mathcal{S}$.*

---

[1]In this paper, we refer to uniform continuity.

The proof is based on the following idea. After showing that $V_\alpha^\star(\mathbf{s})$ is not larger than the optimal value of Problem PMO, we then prove that $V_\alpha^\star(\mathbf{s})$ can only equal to the optimal value of Problem PMO by contradiction. See Appendix B for the proof details. From Theorem 1, we know that the solution of Problem PMO gives the probability measure associated with the action's probability distribution $\boldsymbol{\pi}_\alpha^\star(\cdot|\mathbf{s})$ given by the optimal stationary policy, which is Bellman equation for CCMDPs.

Problem PMO is difficult to solve since we must optimize a probability measure, an infinite-dimensional variable. We further reduce Problem PMO into the following flipping-based policy optimization problem (FPO):

$$\max_{\mathbf{a}_{(1)}, \mathbf{a}_{(2)}, w} \quad w Q_\alpha^\star\left(\mathbf{s}, \mathbf{a}_{(1)}\right) + (1-w) Q_\alpha^\star\left(\mathbf{s}, \mathbf{a}_{(2)}\right) \tag{FPO}$$
$$\text{s.t.} \quad w \mathbb{P}^\star\left(\mathbf{s}, \mathbf{a}_{(1)}\right) + (1-w)\mathbb{P}^\star\left(\mathbf{s}, \mathbf{a}_{(2)}\right) \geq 1 - \alpha.$$

We have the following theorem for Problem FPO:

**Theorem 2.** *Suppose that Assumptions 1 and 2 hold. The optimal objective value of Problem FPO equals to $V_\alpha^\star(\mathbf{s})$ for every $\mathbf{s} \in \mathcal{S}$. Let the solution of Problem FPO be $\mathbf{z}_\alpha^\star(\mathbf{s}) := \left(\mathbf{a}_{(1)}^\star(\mathbf{s}), \mathbf{a}_{(2)}^\star(\mathbf{s}), w^\star(\mathbf{s})\right)$. Define a stationary policy $\tilde{\boldsymbol{\pi}}_\alpha^w(\cdot|\mathbf{s})$ that gives a discrete binary distribution for each $\mathbf{s}$, taking $\mathbf{a} = \mathbf{a}_{(1)}^\star(\mathbf{s})$ with probability $w^\star(\mathbf{s})$ and $\mathbf{a} = \mathbf{a}_{(2)}^\star(\mathbf{s})$ with probability $1 - w^\star(\mathbf{s})$. The policy $\tilde{\boldsymbol{\pi}}_\alpha^w(\cdot|\mathbf{s})$ is an optimal stationary policy with chance constraint, namely, $\tilde{\boldsymbol{\pi}}_\alpha^w(\cdot|\mathbf{s}) \in \Pi_\alpha^\star$.*

The proof is based on the following idea. We first show that Problem PMO has an optimal solution that is a discrete probability measure (Proposition 1 in Appendix C). We then apply the supporting hyperplane theorem and Caratheodory's theorem to show further that the discrete probability measure can be focused on two points. See Appendix C for the proof details. Theorem 2 simplifies the optimizing of the policy in a probability measure space for each $\mathbf{s}$ into an optimization problem in finite-dimensional vector space. An optimal stationary policy gives a discrete binary distribution for each state $\mathbf{s}$. This paper calls the stationary policy with discrete binary distribution as *flipping-based policy* since it is similar to the process of random coin flipping, taking $\mathbf{a} = \mathbf{a}_{(1)}^\star(\mathbf{s})$ with probability $w^\star(\mathbf{s})$ and $\mathbf{a} = \mathbf{a}_{(2)}^\star(\mathbf{s})$ with probability $1 - w^\star(\mathbf{s})$. We summarize one condition that the deterministic policy is enough for the optimality of Problem CCRL in Theorem 3. See Appendix D for the proof.

**Theorem 3.** *Suppose that Assumptions 1 and 2 hold. There exists a deterministic policy that achieves the optimality of Problem CCRL when $\alpha = 0$.*

## 4 Practical Implementation of Flipping-based Policy

This section introduces the practical implementation of flipping-based policy. Obtaining the optimal flipping-based policy for CCMDP is intractable due to the curse of dimensionality [38] and joint chance constraints [43]. The parametrization can tackle the curse of dimensionality. The issue by joint chance constraint is resolved by conservative approximation. The common conservative approximation for joint chance constraint is the linear combination of instantaneous chance constraints. We further show that it is possible to find an expected cumulative safety constraint to conservatively approximate the joint chance constraint, which enables Constrained Policy Optimization (CPO) proposed in [1] to find a conservative approximation of Problem CCRL's optimal flipping-based policy. We show the optimal and finite-sample safety of the flipping-based policy for MDP with the expected cumulative safety constraint.

### 4.1 Extensions to Other Safety Constraints

Except for the joint chance constraints, several other formulations of safety constraints exist, such as expected cumulative [6] and instantaneous constraints [46]. We extend the optimality of flipping-based policy to other safety constraints to show the generality of our result, which may stimulate further study of designing flipping-based policy for other safe RL formulations.

We introduce the extension of the flipping-based policy to MDP with a single expected cumulative safety constraint. The problem is formulated by

$$\max_{\boldsymbol{\pi} \in \Pi} \quad V^{\boldsymbol{\pi}}(\mathbf{s}) \quad \text{s.t.} \quad \mathbb{E}_{\boldsymbol{\tau}_\infty \sim \text{Pr}_{\mathbf{s}_0, \infty}^{\boldsymbol{\pi}}} \left\{ \sum_{i=1}^{\infty} \gamma_{\text{unsafe}}^i \mathbb{I}\left(g(\mathbf{s}_i)\right) | \mathbf{s}_0 = \mathbf{s} \right\} \leq \alpha, \tag{ECRL}$$

where $\gamma_{\text{unsafe}} \in (0, 1)$ is the discount factor and $\mathbb{I}(z)$ defines an indicator function with $\mathbb{I}(z) = 1$ if $z > 0$ and $\mathbb{I}(z) = 0$ otherwise. The following theorem for Problem ECRL holds:

**Theorem 4.** *A flipping-based policy exists in the optimal solution set of Problem* ECRL.

See Appendix E for the proof. The proof follows the same pattern of Theorem 2. We first construct the Bellman recursion with the expected cumulative safety constraint and then prove the existence of a flipping-based policy as optimal policy. The optimality of flipping-based policy can also be extended to the safety constraint function with an additive structure in a finite horizon, written by $\sum_{i=1}^{T} \Pr_{\mathbf{s}, \infty}^{\boldsymbol{\pi}_{\boldsymbol{\theta}}} \{\mathbf{s}_i \in \mathbb{S} \mid \mathbf{s}_0 = \mathbf{s}\}$. This safety constraint refers to affine chance constraints [13]. We summarize the extension to problems with affine chance constraints in Appendix J.

**Remark 1.** *Theorem 4 can be extended to a more general case where the cumulative safety constraint is not limited to an indicator function but can be any Lipschitz continuous function, thereby broadening the applicability of our theory to more practical scenarios.*

## 4.2 Conservative Approximation of Joint Chance Constraint

We resolve the curse of dimensionality by searching for the optimal policy within a set $\Pi_{\boldsymbol{\theta}} \subseteq \Pi$ of parametrized policies with parameters $\boldsymbol{\theta} \in \Theta \subset \mathbb{R}^{n_{\boldsymbol{\theta}}}$, for example, neural networks of a fixed architecture. Here, we use $\boldsymbol{\pi}_{\boldsymbol{\theta}}$ to specify a policy parametrized by $\boldsymbol{\theta}$. If the assumption of the existence of the universal approximator holds, we can approximate the optimal flipping-based policy by using a neural network with state $\mathbf{s}$ as input and $\mathbf{z}_{\alpha}^{\star}(\mathbf{s})$ as output. Another representation of the flipping-based policy is using Gaussian mixture distribution, written by $\boldsymbol{\pi}(\cdot|\mathbf{s}) = w(\mathbf{s})\mathcal{N}(\bar{\mathbf{a}}_{(1)}(\mathbf{s}), \Sigma_{(1)}(\mathbf{s})) + (1 - w(\mathbf{s}))\mathcal{N}(\bar{\mathbf{a}}_{(2)}(\mathbf{s}), \Sigma_{(2)}(\mathbf{s}))$. The output is $\bar{\mathbf{a}}_{(1)}(\mathbf{s}), \bar{\mathbf{a}}_{(2)}(\mathbf{s}), w(\mathbf{s}), \Sigma_{(1)}(\mathbf{s})$, and $\Sigma_{(2)}(\mathbf{s})$.

If we have $w(\mathbf{s}) = w^{\star}(\mathbf{s})$, $\bar{\mathbf{a}}_{(1)}(\mathbf{s}) = \mathbf{a}_{(1)}^{\star}(\mathbf{s})$, and $\bar{\mathbf{a}}_{(2)}(\mathbf{s}) = \mathbf{a}_{(2)}^{\star}(\mathbf{s})$ for every $\mathbf{s}$, the flipping-based policy using Gaussian mixture distribution can approximate the flipping-based policy with binary distribution when the covariances $\Sigma_{(1)}(\mathbf{s})$ and $\Sigma_{(2)}(\mathbf{s})$ vanish for every $\mathbf{s}$ [35]. To simplify the implementation, we can use the neural network that outputs $\mathbf{z}_{\alpha}^{\star}(\mathbf{s})$ and achieve the random search by adding a small Gaussian noise on $\mathbf{a}_{(1)}^{\star}(\mathbf{s})$ and $\mathbf{a}_{(2)}^{\star}(\mathbf{s})$ during implementation.

Rewrite the joint chance constraint (1) with $\mathbf{s}_0 = \mathbf{s}$ for parametrized policy by

$$\Pr_{\mathbf{s}, \infty}^{\boldsymbol{\pi}_{\boldsymbol{\theta}}} \{\mathbf{s}_{k+i} \in \mathbb{S}, \forall i \in [T] \mid \mathbf{s}_k \in \mathbb{S}\} \geq 1 - \alpha, \ \forall k = 0, 1, 2, \dots \quad (3)$$

In local parametrized policy search (LPS) for CCMDPs, $\boldsymbol{\theta}$ is updated by solving

$$\max_{\boldsymbol{\theta} \in \Theta} V^{\boldsymbol{\pi}_{\boldsymbol{\theta}}}(\mathbf{s}) \quad \text{s.t.} \quad (3) \text{ and } D(\boldsymbol{\pi}_{\boldsymbol{\theta}} \| \boldsymbol{\pi}_{\boldsymbol{\theta}_k}) \leq \delta. \quad \text{(LPS)}$$

Here, $D(\cdot)$ denotes a similarity metric between two policies, such as Kullback-Leibler (KL) divergence, and $\delta > 0$ is a step size. The updated policy $\boldsymbol{\pi}_{\boldsymbol{\theta}_{k+1}}$ is parametrized by the solution $\boldsymbol{\theta}_{k+1}$ of Problem LPS. The updating process considers the joint chance constraint. Problem LPS is challenging to solve directly due to joint chance constraint. Since MDP with the expected discounted safety constraint can be solved by the existing safe RL algorithm (e.g. CPO). Thus, by introducing the conservative approximation of joint chance constraint, we enable the existing safe RL algorithm to obtain a conservatively approximate solution of Problem CCRL. First, we introduce the formal definition of the conservative approximation:

**Definition 1.** *A function $C : \Theta \times \mathcal{S} \to \mathbb{R}$ is called a conservative approximation of joint chance constraint (3) if we have*

$$C(\boldsymbol{\theta}, \mathbf{s}) \leq 0, \ \forall \mathbf{s} \in \mathbb{S} \implies \Pr_{\mathbf{s}, \infty}^{\boldsymbol{\pi}_{\boldsymbol{\theta}}} \{\mathbf{s}_{k+i} \in \mathbb{S}, \forall i \in [T] \mid \mathbf{s}_k \in \mathbb{S}\} \geq 1 - \alpha, \ \forall k = 0, 1, 2, \dots \quad (4)$$

Let $H_{\text{unsafe}}(\boldsymbol{\theta}, \mathbf{s}) := \mathbb{E}_{\boldsymbol{\tau}_{\infty} \sim \Pr_{\mathbf{s}_0, \infty}^{\boldsymbol{\pi}_{\boldsymbol{\theta}}}} \{\sum_{i=1}^{\infty} \gamma_{\text{unsafe}}^i \mathbb{I}(g(\mathbf{s}_i)) \mid \mathbf{s}_0 = \mathbf{s}\}$ be a value function for unsafety starting with state $\mathbf{s}$. We have theorem of conservative approximation of joint chance constraint:

**Theorem 5.** *Suppose that $\Theta$ and $\mathbb{S}$ are compact and Assumption 2 holds. Define a function $C_{\text{unsafe}}(\boldsymbol{\theta}, \mathbf{s})$ by $C_{\text{unsafe}}(\boldsymbol{\theta}, \mathbf{s}) := H_{\text{unsafe}}(\boldsymbol{\theta}, \mathbf{s}) - \alpha$. There exist large enough $\gamma_{\text{unsafe}}$ and small enough $T$ such that $C_{\text{unsafe}}(\boldsymbol{\theta}, \mathbf{s}) \leq 0$ is a conservative approximation of (3).*

The proof of Theorem 5 is summarized in Appendix F. We formulate a conservative approximation of Problem LPS (CLPS) as follows:

$$\max_{\boldsymbol{\theta} \in \Theta} V^{\boldsymbol{\pi}_{\boldsymbol{\theta}}}(\mathbf{s}) \quad \text{s.t.} \quad H_{\text{unsafe}}(\boldsymbol{\theta}, \mathbf{s}) \leq \alpha, \ D(\boldsymbol{\pi}_{\boldsymbol{\theta}} \| \boldsymbol{\pi}_{\boldsymbol{\theta}_k}) \leq \delta. \quad \text{(CLPS)}$$

By Theorem 5, the optimal solution of Problem CLPS is a feasible solution of Problem LPS and thus the corresponding parametric policy is within $\Pi_\alpha^\star$. We have a remark on Theorem 5 as follows:

**Remark 2.** *By the same procedure of proving Theorem 5, we can show that Problem* ECRL *is a conservative approximation of Problem* CCRL.

### 4.3 Practical Algorithms

Then, we present a practical way to train the optimal flipping-based policy using existing tools in the infrastructural frameworks for safe reinforcement learning research, such as OmniSafe [24]. The provided tools can train the deterministic or Gaussian distribution-based stochastic policies. We take the parameterized deterministic policies as an example, which is specified by $\pi_{\boldsymbol{\theta}}^{\mathsf{d}}$. The parameter $\boldsymbol{\theta}$ is within a compact set $\boldsymbol{\Theta}$. We write the reinforcement learning of parameterized deterministic policy (PDPRL) for CCMDPs by

$$\max_{\boldsymbol{\theta} \in \boldsymbol{\Theta}} J(\boldsymbol{\theta}) := \mathbb{E}_{\boldsymbol{\tau}_\infty \sim \pi_{\boldsymbol{\theta}}^{\mathsf{d}}} \left\{ R(\boldsymbol{\tau}_\infty) \right\} \quad \text{s.t.} \quad F^{\mathsf{d}}(\boldsymbol{\theta}) \geq 1 - \alpha. \tag{PDPRL}$$

Here, the constraint function $F^{\mathsf{d}}(\boldsymbol{\theta})$ is defined by

$$F^{\mathsf{d}}(\boldsymbol{\theta}) := \mathbb{E}_{\mathbf{s}_0 \sim \mu_0} \left\{ \mathrm{Pr}_{\mathbf{s},\infty}^{\pi_{\boldsymbol{\theta}}^{\mathsf{d}}} \left\{ \mathbf{s}_i \in \mathbb{S}, \forall i \in [T] | \mathbf{s}_0 \right\} \right\}.$$

Let $J_\alpha^*$ and $\boldsymbol{\Theta}_\alpha^*$ be the optimal value and optimal solution set of Problem PDPRL. Different from the previous discussions, we here consider the expectation of the initial state $\mathbf{s}_0$ instead of considering the problem for each $\mathbf{s}_0$. The reason is that the provided tools in OmniSafe, for example, CPO [1] and PCPO [48], address the problems in which the reward functions consider the expectation of the initial state. We extend the previous results of the flipping-based policy to this case.

Let $\mathscr{B}(\boldsymbol{\Theta})$ be the Borel $\sigma$-algebra ($\sigma$-field) on $\boldsymbol{\Theta} \subset \mathbb{R}^{n_\theta}$ with Euclidean distance. Let $\nu \in M(\boldsymbol{\Theta})$ be a probability measure on $(\boldsymbol{\Theta}, \mathscr{B}(\boldsymbol{\Theta}))$. With the above notation, associated with Problem PDPRL, a reinforcement learning of parameterized stochastic policy (PSPRL) is formulated as:

$$\max_{\nu \in M(\boldsymbol{\Theta})} \int_{\boldsymbol{\Theta}} J(\boldsymbol{\theta}) \mathrm{d}\nu \quad \text{s.t.} \quad \int_{\boldsymbol{\Theta}} F^{\mathsf{d}}(\boldsymbol{\theta}) \mathrm{d}\nu \geq 1 - \alpha. \tag{PSPRL}$$

Let $M_\alpha(\boldsymbol{\Theta}) := \left\{ \mu \in M(\boldsymbol{\Theta}) : \int_{\boldsymbol{\Theta}} F^{\mathsf{d}}(\boldsymbol{\theta}) \mathrm{d}\nu \geq 1 - \alpha \right\}$ be the feasible set of Problem PSPRL. The optimal objective value and the optimal solution set of Problem PSPRL are $\mathcal{J}_\alpha^* := \max_{\nu \in M_\alpha(\boldsymbol{\Theta})} \int_{\boldsymbol{\Theta}} J(\boldsymbol{\theta}) \mathrm{d}\nu$ and $A_\alpha := \left\{ \mu \in M_\alpha(\boldsymbol{\Theta}) : \int_{\boldsymbol{\Theta}} J(\boldsymbol{\theta}) \mathrm{d}\nu = \mathcal{J}_\alpha^* \right\}$. A probability measure $\nu_\alpha^* \in A_\alpha$ is called an optimal probability measure for Problem PSPRL. Define $\mathbf{V}_{\mathsf{m}} := \left\{ \nu_{\mathsf{m}} \in [0,1]^2 : \sum_{i=1}^2 \nu_{\mathsf{m}}(i) = 1 \right\}$. Let $\boldsymbol{\zeta}_{\mathsf{m}} := \left( \nu_{\mathsf{m}}(1), \nu_{\mathsf{m}}(2), \boldsymbol{\theta}^{(1)}, \boldsymbol{\theta}^{(2)} \right)$ be a variable in the set $\mathbf{Z}_{\mathsf{m}} := \mathbf{V}_{\mathsf{m}} \times \boldsymbol{\Theta}^2$. Consider an optimization problem on $\boldsymbol{\zeta}_{\mathsf{m}}$, reinforcement learning of parameterized flipping-based policy (PFPRL), written as

$$\max_{\boldsymbol{\zeta}_{\mathsf{m}} \in \mathbf{Z}_{\mathsf{m}}} \sum_{i=1}^2 J(\boldsymbol{\theta}^{(i)}) \nu_{\mathsf{m}}(i) \quad \text{s.t.} \quad \sum_{i=1}^2 \nu_{\mathsf{m}}(i) F^{\mathsf{d}}(\boldsymbol{\theta}^{(i)}) \geq 1 - \alpha. \tag{PFPRL}$$

Define $\mathbf{Z}_{\mathsf{m},\alpha} := \left\{ \boldsymbol{\zeta}_{\mathsf{m}} \in \mathbf{Z}_{\mathsf{m}} : \sum_{i=1}^2 F^{\mathsf{d}}(\boldsymbol{\theta}^{(i)}) \nu_{\mathsf{m}}(i) \geq 1 - \alpha \right\}$ as the feasible set of Problem PFPRL. In addition, define the optimal objective value and optimal solution set of Problem PFPRL by $\mathcal{J}_\alpha^{\mathsf{w}} := \min \left\{ \sum_{i=1}^2 J(\boldsymbol{\theta}^{(i)}) \nu_{\mathsf{m}}(i) : \boldsymbol{\zeta}_{\mathsf{m}} \in \mathbf{Z}_{\mathsf{m},\alpha} \right\}$ and $D_\alpha := \left\{ \boldsymbol{\zeta}_{\mathsf{m}} \in \mathbf{Z}_{\mathsf{m},\alpha} : \sum_{i=1}^2 J(\boldsymbol{\theta}^{(i)}) \nu_{\mathsf{m}}(i) = \mathcal{J}_\alpha^{\mathsf{w}} \right\}$, respectively. We have Theorem 6 for parameterized flipping-based policy.

**Theorem 6.** *The optimal values of Problems* PFPRL *and* PSPRL *are equal,* $\mathcal{J}_\alpha^* = \mathcal{J}_\alpha^{\mathsf{w}}$. *If* $J_\alpha^*$ *is a strictly convex function of* $\alpha$ *on an interval* $(\underline{\alpha}, \overline{\alpha})$, *then,* $\forall \alpha \in (\underline{\alpha}, \overline{\alpha})$, $\mathcal{J}_\alpha^* > J_\alpha^*$ *holds.*

See Appendix G for the proof. Note that Theorem 6 clarifies the existence of a parameterized flipping-based policy achieving optimality and the condition under which it performs better than deterministic policies.

**Remark 3.** *Theorems 6 and 2 have different results on the flipping probability. Theorem 2 claims a state-dependent flipping probability while the flipping probability in Theorem 6 is fixed. The*

---

**Algorithm 1** General training algorithm for flipping-based policy

---

1: Set a positive integer $S \in \mathbb{N}_+$ and obtain the sample set $\mathcal{Z}_S = \{\tilde{\alpha}_i\}_{i=1}^S$ by randomly sampling $\tilde{\alpha}_i \in [0, 1], \forall i \in [S]$ according to uniform distribution
2: Optimize a policy parameter $\tilde{\boldsymbol{\theta}}_i$ for each $\tilde{\alpha}_i$ via an existing safe RL algorithm
3: Solving linear program LP and obtain $\left( \nu_\mathsf{s}(j_1^*), \nu_\mathsf{s}(j_2^*), \tilde{\boldsymbol{\theta}}_{j_1^*}, \tilde{\boldsymbol{\theta}}_{j_2^*} \right)$

---

---

**Algorithm 2** Flipping-based policy implementation

---

1: Observe the state $\mathbf{s}_k$ at time step $k = 0, 1, 2, ...$
2: Randomly generate a number $\kappa$ from $[0, 1]$ obeying uniform distribution
3: If $\kappa \leq \nu_\mathsf{s}(j_1^*)$, generate $\mathbf{a}_k$ by $\boldsymbol{\pi}_{\tilde{\boldsymbol{\theta}}_{j_1^*}}^\mathsf{d}$. Otherwise, generate $\mathbf{a}_k$ implement $\boldsymbol{\pi}_{\tilde{\boldsymbol{\theta}}_{j_2^*}}^\mathsf{d}$

---

*distinction arises from a subtle difference between Problem* CCRL *and Problem* PSPRL*. In Problem* CCRL*, the optimal policy for each state is derived based on a revised Bellman equation, ensuring that the joint chance constraint is satisfied pointwise for every state. On the other hand, Problem* PSPRL *focuses on the expectation of the joint chance constraint, evaluated over the probability distribution of the initial state. This formulation eliminates the need for pointwise satisfaction across the state space, causing the state-dependent nature of the constraint to disappear.*

There is no existing tool to solve Problem PFPRL and we can only apply them to solve Problem PDPRL for any given $\alpha$. Let $\mathcal{Z}_S = \{\tilde{\alpha}_i\}_{i=1}^S$, $\tilde{\alpha}_i \in [0, 1]$, $\forall i \in [S]$ be a set of probability levels. For each $\tilde{\alpha}_i$, define $\tilde{\boldsymbol{\theta}}_i$ be the optimal solution of Problem PDPRL with $\alpha = \tilde{\alpha}_i$. Consider the linear program (LP):

$$\max_{\nu_\mathsf{s}(1),...,\nu_\mathsf{s}(S) \in [0,1]^S} \sum_{i=1}^S J(\tilde{\boldsymbol{\theta}}_i) \nu_\mathsf{s}(i) \quad \text{s.t.} \quad \sum_{i=1}^S \nu_\mathsf{s}(i) F^\mathsf{d}(\tilde{\boldsymbol{\theta}}_i) \geq 1 - \alpha, \ \sum_{i=1}^S \nu_\mathsf{s}(i) = 1. \quad \text{(LP)}$$

Define the optimal objective value and optimal solution set $\widetilde{D}_\alpha(\mathcal{Z}_S)$ of Problem LP by $\widetilde{\mathcal{J}}_\alpha^\mathsf{k}(\mathcal{Z}_S)$ and $\widetilde{D}_\alpha(\mathcal{Z}_S)$, respectively. The optimal flipping-based policy is characterized by $\left( \nu_\mathsf{s}(j_1^*), \nu_\mathsf{s}(j_2^*), \tilde{\boldsymbol{\theta}}_{j_1^*}, \tilde{\boldsymbol{\theta}}_{j_2^*} \right)$, where $j_1^*$ and $j_2^*$ are the index for the non-zero elements of the optimal solution of linear program LP The following theorem holds for $\widetilde{\mathcal{J}}_\alpha^\mathsf{k}(\mathcal{Z}_S)$ and $\widetilde{D}_\alpha(\mathcal{Z}_S)$.

**Theorem 7.** *There exists an optimal solution in $\widetilde{D}_\alpha(\mathcal{Z}_S)$ such that the number of non-zero elements does not exceed two. Besides, if $\tilde{\alpha}_i$ is extracted independently and identically (uniform distribution), as $S \to \infty$, we have $\widetilde{\mathcal{J}}_\alpha^\mathsf{k}(\mathcal{Z}_S) \to \mathcal{J}_\alpha^*$ with probability 1.*

See Appendix H for the proof. Theorem 6 shows that there exists an optimal solution to Problem PSPRL that is a linear combination of two deterministic policies. Theorem 7 clarifies that we could obtain an approximate flipping-based policy to Problem PSPRL by optimizing the linear combination of multiple trained optimal deterministic policies. One is the linear combination of two policies among all possible linear combinations. The above conclusions can be extended to the Gaussian distribution-based stochastic policies. Besides, **the above conclusions still hold after replacing the chance constraint with the expected cumulative safe constraint in CPO and PCPO.** We summarize a general algorithm for approximately training the flipping-based policy based on the existing safe RL algorithms in Algorithm 1. With $\left( \nu_\mathsf{s}(j_1^*), \nu_\mathsf{s}(j_2^*), \tilde{\boldsymbol{\theta}}_{j_1^*}, \tilde{\boldsymbol{\theta}}_{j_2^*} \right)$, we implement the flipping-based policy by Algorithm 2. In the practical implementation, the weight is constant instead of a function of the initial state since Problems PDPRL and PSPRL consider the expectation of the initial state. Besides, $\tilde{\alpha}_i$ in (1) is replaced by cost limit when using CPO or PCPO to obtain a conservative approximation of (3).

## 4.4 Safety with Finite Samples

The update by solving Problem CLPS is difficult to implement practically since the evaluation of the constraint function $H_\mathsf{unsafe}(\boldsymbol{\theta}, \mathbf{s}) \leq \alpha$ is necessary to clarify whether a policy $\boldsymbol{\pi}$ is feasible,

which is challenging in high-dimensional cases. Here, we apply the surrogate functions proposed in [1] to replace the objective and constraints of Problem CLPS. With a probability $\alpha_{\mathsf{s}} \in [0, \alpha)$, the CPO-based approximation of Problem CLPS (CPOS) is written by

$$
\max_{\boldsymbol{\theta} \in \Theta} \mathbb{E}_{\mathbf{s}_{\mathsf{ini}} \sim \boldsymbol{\pi}_{\boldsymbol{\theta}_k}, \mathbf{a} \sim \boldsymbol{\pi}_{\boldsymbol{\theta}}} \left\{ r(\mathbf{s}_{\mathsf{ini}}, \mathbf{a}) \right\}
$$
$$
\text{s.t. } H_{\mathsf{unsafe}}(\boldsymbol{\theta}_k, \mathbf{s}) + \frac{1}{1 - \gamma_{\mathsf{unsafe}}} \mathbb{E}^{\mathbf{a} \sim \boldsymbol{\pi}_{\boldsymbol{\theta}}}_{\mathbf{s}_{\mathsf{ini}} \sim \boldsymbol{\pi}_{\boldsymbol{\theta}_k}} \left\{ \mathbb{I}(g(\mathbf{s}^+)) \right\} \le \alpha_{\mathsf{s}}, \ D(\boldsymbol{\pi}_{\boldsymbol{\theta}} \| \boldsymbol{\pi}_{\boldsymbol{\theta}_k}) \le \delta. \tag{CPOS}
$$

Note that the optimal solution $\boldsymbol{\theta}_{k+1}$ of Problem CPOS may differ from the one of Problem CLPS. Proposition 2 of [1] gives the upper bound of CPO update constraint violation. The upper bound depends on the values of the step size $\delta$, probability level $\alpha_{\mathsf{s}}$, discount factor $\gamma_{\mathsf{unsafe}}$, and the maximal expected risk defined by $\eta^{\boldsymbol{\theta}_{k+1}}_{\alpha_{\mathsf{s}}} := \max_{\mathbf{s}_{\mathsf{ini}} \in \mathbb{S}} \mathbb{E}_{\mathbf{a} \sim \boldsymbol{\pi}_{\boldsymbol{\theta}_{k+1}}} \left\{ \mathbb{I}(g(\mathbf{s}^+)) \right\}$. The upper bound can be written by $H_{\mathsf{unsafe}}(\boldsymbol{\theta}_{k+1}, \mathbf{s}) \le \alpha_{\mathsf{s}} + \frac{\sqrt{2\delta} \gamma_{\mathsf{unsafe}} \eta^{\boldsymbol{\theta}_{k+1}}_{\alpha_{\mathsf{s}}}}{(1 - \gamma_{\mathsf{unsafe}})^2}$. By choosing sufficiently small step size $\delta$, discount factor $\gamma_{\mathsf{unsafe}}$, and probability level $\alpha_{\mathsf{s}}$, it is able to ensure that $H_{\mathsf{unsafe}}(\boldsymbol{\theta}_{k+1}, \mathbf{s}) \le \alpha$.

In practical implementation, the exact value of $\mathbb{E}_{\mathbf{s} \sim \boldsymbol{\pi}_{\boldsymbol{\theta}_k}, \mathbf{a} \sim \boldsymbol{\pi}_{\boldsymbol{\theta}}} \left\{ \mathbb{I}(g(\mathbf{s}^+)) \right\}$ is unavailable and samples of $\mathbf{s}_{\mathsf{ini}} \sim \boldsymbol{\pi}_{\boldsymbol{\theta}_k}$ and $\mathbf{a} \sim \boldsymbol{\pi}_{\boldsymbol{\theta}}$ are used to approximate the CPO update. The data set is defined by $\mathcal{D}_N := \left\{ (\mathbf{s}^{(i)}, \mathbf{a}^{(i)}, \mathbf{s}^{+,(i)}) \right\}^N_{i=1}$, where $N \in \mathbb{N}_+$ is the sample number and $\mathbf{s}^{+,(i)}$ is a sample of successor with previous state $\mathbf{s}^{(i)}$ and action $\mathbf{a}^{(i)}$. Instead of directly solving Problem CPOS, the following sample average approximate of Problem CPOS (S-CPOS) is solved:

$$
\max_{\boldsymbol{\theta} \in \Theta} \frac{1}{N} \sum_{i=1}^N r(\mathbf{s}^{(i)}_{\mathsf{ini}}, \mathbf{a}^{(i)}) \quad \text{s.t.} \quad \widetilde{H}^{\mathsf{loc}}_{\mathsf{unsafe}}(\boldsymbol{\theta}_k, \mathbf{s}, \gamma_{\mathsf{unsafe}}, \mathcal{D}_N) \le \tilde{\alpha}_{\mathsf{s}}, \ D(\boldsymbol{\pi}_{\boldsymbol{\theta}} \| \boldsymbol{\pi}_{\boldsymbol{\theta}_k}) \le \delta. \tag{S-CPOS}
$$

Here, $\widetilde{H}^{\mathsf{loc}}_{\mathsf{unsafe}}(\boldsymbol{\theta}_k, \mathbf{s}, \gamma_{\mathsf{unsafe}}, \mathcal{D}_N) := H_{\mathsf{unsafe}}(\boldsymbol{\theta}_k, \mathbf{s}) + \frac{1}{(1 - \gamma_{\mathsf{unsafe}})N} \sum_{i=1}^N \mathbb{I}(g(\mathbf{s}^{+,(i)}))$ and $\tilde{\alpha}_{\mathsf{s}} \in [0, \alpha_{\mathsf{s}})$ is a probability level. The extraction of sample set $\mathcal{D}_N$ is random, and thus the optimal solution $\tilde{\boldsymbol{\theta}}_{\tilde{\alpha}_{\mathsf{s}}}(\mathbf{s}, \boldsymbol{\theta}_k, \mathcal{D}_N)$ of Problem S-CPOS is a random variable due to the independence on the sample set $\mathcal{D}_N$. We need to investigate the probability that $\tilde{\boldsymbol{\theta}}_{\tilde{\alpha}_{\mathsf{s}}}(\mathbf{s}, \boldsymbol{\theta}_k, \mathcal{D}_N)$ admits a feasible policy for Problem CCRL. We have Theorem 8 for the safety with finite sample number. See Appendix I for the proof.

**Theorem 8.** *Suppose that the step size $\delta > 0$ and $\alpha_{\mathsf{s}} \in [0, \alpha)$ are adjusted to ensure that $H_{\mathsf{unsafe}}(\boldsymbol{\theta}_{k+1}, \mathbf{s}) \le \alpha$. There exist $\overline{T}$ and $\underline{\gamma}_{\mathsf{unsafe}}$ such that, if $\tilde{\alpha}_{\mathsf{s}} \in [0, \alpha_{\mathsf{s}})$, $\gamma_{\mathsf{unsafe}} > \underline{\gamma}_{\mathsf{unsafe}}$, and $T < \overline{T}$, such that $\tilde{\boldsymbol{\theta}}_{\tilde{\alpha}_{\mathsf{s}}}(\mathbf{s}, \boldsymbol{\theta}_k, \mathcal{D}_N)$ admits a feasible policy for Problem CCRL with a probability larger than $1 - \exp\left\{ -2N(\alpha_{\mathsf{s}} - \tilde{\alpha}_{\mathsf{s}})^2 (1 - \gamma_{\mathsf{unsafe}})^2 \right\}$.*

Note that Theorems 5 and 8 only show the existence of the parameters for safety but do not show an explicit way to choose $\gamma_{\mathsf{unsafe}}$ for specified $T$. If a conservative safety is desired for practical applications, we recommend using a $\gamma_{\mathsf{unsafe}}$ close to 1.

## 5 Experiments

### 5.1 Numerical Example

We conduct a numerical example to illustrate how the flipping-based policy outperforms the deterministic policy in CCMDPs. The numerical example considers driving a point from the initial point $(0, 0)$ to the goal $(15, 15)$ with the probability of entering dangerous regions smaller than a required value. The uncertainties come from the disturbances to the implemented actions. The metric for evaluating the performance is the cumulative inverse distance to the goal, a reward function. Due to the page limit, we summarize the details of the model and heuristic method for obtaining the neural network-based policy in Appendix K. Figure 2 (a) shows trajectories by the deterministic policy in one thousand simulations with mean reward as 0.8667 and violation probability as 17%. The red trajectories have intersections with the dangerous region. The deterministic policy led to a sideway in front of the dangerous regions since crossing the middle space violates the violation probability constraint. Figure 2 (b) shows trajectories by flipping-based policy. The mean reward was reduced to 1.8259 while the violation probability is 17%, the same as the deterministic policy. The reason is that the flipping-based policy sometimes took the risk of crossing the middle space to improve the mean reward. To balance the violation probability, the sideway root taken by the flipping-based policy was

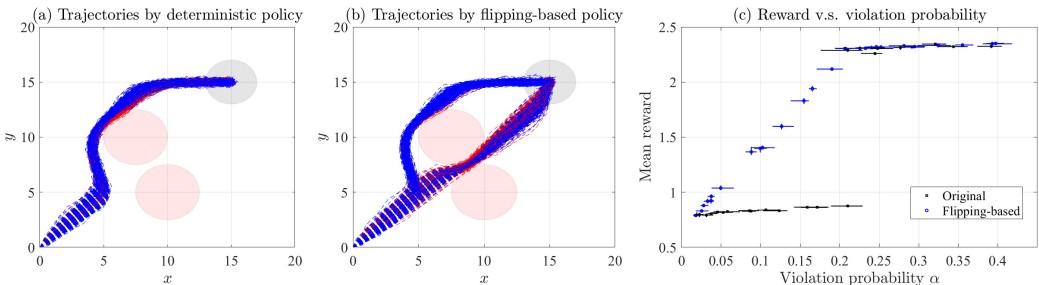

Figure 2: Results on the numerical example. Blue dashed lines are feasible trajectories that reach the goal set (grey shaded circle) and avoid dangerous regions (red shaded circles)). Red dashed lines mean that the constraint of avoiding dangerous regions is violated. (a) Trajectories by the deterministic policy with $\alpha = 17\%$. The mean reward is $0.8667$; (b) Trajectories by the flipping-based policy with $\alpha = 17\%$. The mean reward is $1.8259$; (c) Profile of the mean reward along with the violation probability. Error bars represent the minimal and maximal values across five different simulation sets.

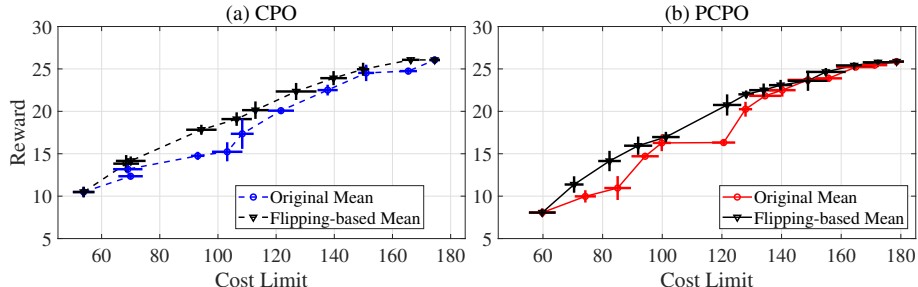

Figure 3: Experimental results on Safety Gym (PointGoal2). Adopting the flipping-based policy increases the expected reward under the same expected cost for CPO and PCPO at intervals where the reward profile is convex. Error bars represent $1\sigma$ confidence intervals across five different random seeds.

more conservative than the deterministic policy. Figure 2 (c) gives the profile of the mean reward along with the violation probability. Until around $22\%$, the flipping-based policy outperformed the deterministic policy since the violation probability of crossing the middle space is larger than that, and the deterministic policy can cross it. The profile in Figure 2 has a convex shape until $22\%$. Theorem 6 points out that the strict convexity implies the better reward performance of the flipping-based policy.

## 5.2 Safety Gym

We conduct experiments on Safety Gym [34], where an agent must maximize the expected cumulative reward under a safety constraint with additive structures. The reason for choosing Safety Gym is that this benchmark is complex and elaborate and has been used to evaluate various excellent algorithms. The infrastructural framework for performing safe RL algorithms is OmniSafe [24]. The proposed method has been validated in two environments: PointGoal2 and CarGoal2. Four algorithms are used as baselines. The first is CPO [1], a well-known algorithm for solving CMDPs. The other three algorithms are PCPO [48], P3O [50], and CUP [47], recent algorithms that achieve superior performance compared to CPO. Due to space limitations, we only present the experimental results of the test processes for CPO and PCPO on PointGoal2. The details of the experimental setup, all four algorithms' training process results on PointGoal2 and their test process results on CarGoal2 are provided in Appendix L.

**Baselines and metrics.** We implement the practical algorithms presented in Section 4.3 to obtain the flipping-based policy. We modified Algorithm 1 to train the flipping-based policy based on CPO and PCPO. In Algorithm 1, the sample set $\mathcal{Z}_S$ consists of samples of violation probabilities. Since CPO and PCPO consider the expected cumulative safety constraints, the sample set $\mathcal{Z}_S$ includes the samples of cost limits of the expected cumulative safety constraints. Instead of using the training

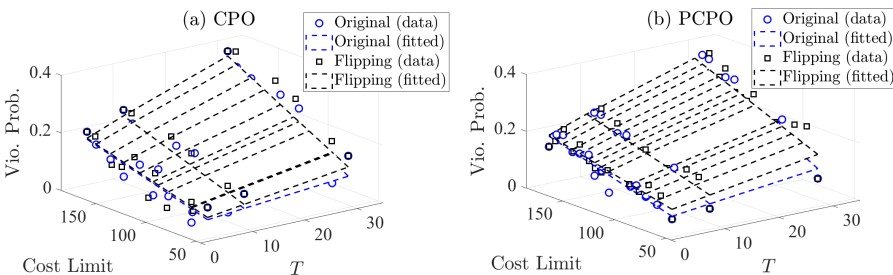

Figure 4: Experimental results on Safety Gym (PointGoal2). The relationship between expected cumulative safety and violation probabilities.

process, we compared the performance of the testing process, where we implemented the trained policy with new randomly generated initial states and goal points and evaluated the expectations of the reward and cost for each trained policy. We investigate whether the expected reward of each baseline under the same expected cost limit can be improved by transforming the policy into a flipping-based policy without any other changes. We employ the expected cumulative reward and the expected cumulative safety as metrics to evaluate the flipping-based policy and the aforementioned baselines. We execute CPO and PCPO with five random seeds and compute the means and confidence intervals.

**Results.** The experimental results are summarized in Figure 3. As shown in the figure, for CPO and PCPO, the expected reward increases as the expected cost limit rises, and it exhibits convexity at some intervals. At intervals with convexity, the flipping-based policy significantly increases the expected reward. While at intervals without convexity, the flipping-based policy does not increase the expected reward. The above observation fits Theorem 6. From the experiments including more details in Appendix L, we also observe that our flipping-based policy can generally enhance existing safe RL algorithms, although the degree of improvement depends on the original algorithm's performance. Essentially, the flipping mechanism is a linear combination of a performance policy (risky but high-performing) and a safety policy (safe but lower-performing) designed to increase the reward while maintaining the required level of risk. One concern regarding the results in Figure 3 is that the flipping-based policy introduces broader confidence intervals. In theory, however, the flipping-based policy does not increase the size of the confidence intervals. This is demonstrated in the numerical example in Section 5.1, where the policy achieves solutions closer to the optimal ones as outlined in Theorem 2. The practical implementation described in Section 4.3, however, may experience broader confidence intervals due to the presence of two sources of Gaussian noise. It is possible to mitigate this issue by reducing the degree of stochasticity in the policy, for instance, by using smaller variances for the Gaussian noise. This adjustment would not negatively impact the performance in terms of mean reward and cost.

On the other hand, we summarize the results of the relation between expected cumulative safety and the violation probability in Figure 4. Expected cumulative safety and violation probability follows a linear causality, indicating that the flipping-based policy outperforms the deterministic policy under joint chance constraint. With the same expected cumulative safety, a larger $T$ introduces a larger violation probability, which validates Theorem 5.

## 6  Conclusions

In this article, we first introduce the Bellman equation for CCMDP and prove that a flipping-based polity exists that achieves optimality. We then proposed practical implementation of approximately training the flipping-based policy for CCMDP. Conservative approximations of joint chance constraints were presented. Specifically, we introduced a framework for adapting Constrained Policy Optimization (CPO) to train a flipping-based policy. This framework can be easily adapted to other safe RL algorithms. Finally, we demonstrated that a flipping-based policy can improve the performance of safe RL algorithms under the same safety constraints limits on the Safety Gym benchmark.

## Acknowledgements and Disclosure of Funding

We would like to thank the anonymous reviewers for their helpful comments. This work is partially supported by LY Corporation, JSPS Kakenhi (24K16752), Research Organization of Information and Systems via 2023-SRP-06, Osaka University Institute for Datability Science (IDS interdisciplinary Collaboration Project), JST CREST JPMJCR201, and NFR project SARLEM.

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

# Appendix

## A  Limitations and Potential Negative Societal Impacts

**Limitations.**  The practical algorithm (Algorithm 1) for training the flipping-based policy is adaptable to any safe RL algorithms. However, it has the following limitations. First, there is a gap in the scenarios with non-smooth functions. The results should be extended to more practical scenarios with non-smooth functions. Second, training a couple of policies is required to find an optimal combination for each cost limit. We could not figure out the convexity after getting enough pairs of expected rewards and costs. Third, the probability of the flip in the practical algorithm is not state-dependent. Although it achieves the optimality of the parameterized flipping-based policy when considering the expectation of the initial state, optimality by Theorem 2 has not yet been achieved. Future work should focus on designing a practical algorithm to obtain the flipping-based policy, for example, training a neural network to take action candidates and the probability of flip as output and the state as input. If we could develop an efficient algorithm to learn the flipping-based policy given by Theorem 2, there is no need to consider the tradeoff between performance and computational complexity.

**Potential negative societal impacts.**  We believe that safety is an essential requirement for applying reinforcement learning (RL) in many real-world problems. While we have not identified any potential negative societal impacts of our proposed method, we must acknowledge that RL algorithms, including ours, are vulnerable to misuse. It is crucial to remain vigilant about the ethical implications and potential risks associated with their application.

## B  Proof of Theorem 1

The proof sketch is summarized as follows:

(a) Show that $V_\alpha^\star(\mathbf{s})$ is not larger than the optimal value of Problem PMO;

(b) Assume that $V_\alpha^\star(\mathbf{s})$ is smaller than the optimal value of Problem PMO;

(c) From (b), we can construct a better policy than which implies that $V_\alpha^\star(\mathbf{s})$ is not the optimal value function;

(d) Since (c) contradicts the fact that $V_\alpha^\star(\mathbf{s})$ is the optimal value function, $V_\alpha^\star(\mathbf{s})$ cannot be smaller than the optimal value of Problem PMO and only equality holds. From the equality, we prove Theorem 1.

Following the above sketch, the proof of Theorem 1 is as follows:

*Proof of Theorem 1.*  For a state $\mathbf{s}$, let $\boldsymbol{\mu}_\alpha^* \in M(\mathcal{A})$ be the solution of Problem PMO and the associated probability density function is $p_\alpha^*(\cdot)$. Note that $\boldsymbol{\pi}_\alpha^\star$ is a stationary policy and $\boldsymbol{\pi}_\alpha^\star \in \Pi_\alpha$. Thus, the probability measure associated with $\boldsymbol{\pi}_\alpha^\star(\cdot|\mathbf{s})$ is a feasible solution of Problem PMO. By the definitions of $V_\alpha^\star(\mathbf{s})$ and $Q_\alpha^\star(\mathbf{s}, \mathbf{a})$, we have

$$V_\alpha^\star(\mathbf{s}) = \mathbb{E}_{\mathbf{a} \sim \boldsymbol{\pi}_\alpha^\star} \left\{ r(\mathbf{s}, \mathbf{a}) + \gamma \mathbb{E}_{\boldsymbol{\tau}_\infty \sim \mathrm{Pr}_{\mathbf{s}_0, \infty}^{\boldsymbol{\pi}}} \left\{ V_\alpha^\star(\mathbf{s}^+) \,|\mathbf{s}_0 = \mathbf{s}, \mathbf{a}_0 = \mathbf{a} \right\} \right\}$$
$$= \mathbb{E}_{\mathbf{a} \sim \boldsymbol{\pi}_\alpha^\star} \{ Q_\alpha^\star(\mathbf{s}, \mathbf{a}) \} \leq \mathbb{E}_{\mathbf{a} \sim p_\alpha^*} \{ Q_\alpha^\star(\mathbf{s}, \mathbf{a}) \} .$$

Suppose $V_\alpha^\star(\mathbf{s}) < \mathbb{E}_{\mathbf{a} \sim p_\alpha^*} \{ Q_\alpha^\star(\mathbf{s}, \mathbf{a}) \}$. Since $\boldsymbol{\mu}_\alpha^*$ is a feasible solution of Problem PMO, we have

$$\int_\mathcal{A} \mathbb{P}^\star(\mathbf{s}, \mathbf{a}) \, d\boldsymbol{\mu}_\alpha^* \geq 1 - \alpha. \tag{5}$$

Thus, by implementing $\boldsymbol{\mu}_\alpha^*$ when the state is $\mathbf{s}$, the probability of having $\mathbf{s}_{k+i} \in \mathbb{S}, \forall i \in [L]$ is larger than $1 - \alpha$.

Construct a new policy $\tilde{\boldsymbol{\pi}}_\alpha^*$ by

- The state is $\mathbf{s}$: $\tilde{\boldsymbol{\pi}}_\alpha^* = p_\alpha^*(\cdot)$;

- The state is not $\mathbf{s}$: $\tilde{\boldsymbol{\pi}}_\alpha^* = \boldsymbol{\pi}_\alpha^\star$.

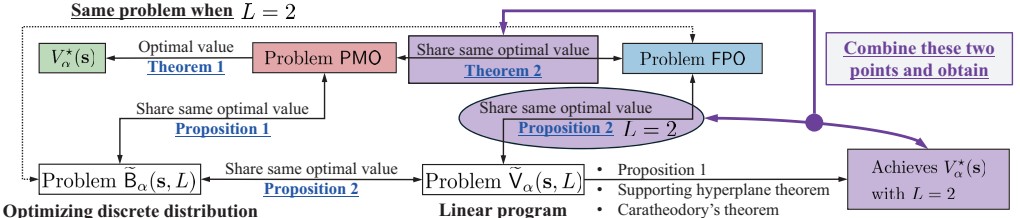

Figure 5: Proof sketch of Theorem 2.

Due to (5) and $\boldsymbol{\pi}_\alpha^\star \in \Pi_\alpha$, we have that $\tilde{\boldsymbol{\pi}}_\alpha^* \in \Pi_\alpha$ since $\tilde{\boldsymbol{\pi}}_\alpha^*$ satisfies the chance constraint (1). Therefore, we have

$$V^{\tilde{\boldsymbol{\pi}}_\alpha^*}(\mathbf{s}) = \mathbb{E}_{\mathbf{a}\sim\tilde{\boldsymbol{\pi}}_\alpha^*}\left\{Q^{\tilde{\boldsymbol{\pi}}_\alpha^*}(\mathbf{s},\mathbf{a})\right\} = \mathbb{E}_{\mathbf{a}\sim p_\alpha^*}\left\{Q_\alpha^\star(\mathbf{s},\mathbf{a})\right\} > V_\alpha^\star(\mathbf{s}). \tag{6}$$

Note that (6) contradicts with the fact that $\boldsymbol{\pi}_\alpha^\star$ is the optimal stationary policy, which completes the proof. $\qquad\square$

## C  Proof of Theorem 2

To help understand the proof of Theorem 2, we illustrate the proof sketch by Figure 5.

Let $L \in \mathbb{N}_+$ be a positive integer and $[L] := \{1, ..., L\}$ be the set of the index. Consider the augmented space $\mathcal{A}^L$ and define an element of $\mathcal{S}^L$ by $\mathcal{C}_L = \left(\mathbf{a}_{(1)}, ..., \mathbf{a}_{(L)}\right)$. For an arbitrarily given $\mathcal{C}_L$, we define a set of discrete probability measures by

$$\mathcal{U}_{\mathsf{d},L} := \left\{\boldsymbol{\mu}_{\mathsf{d},L} \in [0,1]^L : \sum_{i=1}^L \boldsymbol{\mu}_{\mathsf{d},L}(i) = 1\right\}. \tag{7}$$

The set $\mathcal{C}_L$ becomes a sample space with finite samples if it is equipped with a discrete probability measure $\boldsymbol{\mu}_{\mathsf{d},L} \in \mathcal{U}_{\mathsf{d},L}$, where the $i$-th element $\boldsymbol{\mu}_{\mathsf{d},L}(i)$ denotes the probability of taking decision $\mathbf{a}_{(i)}$, i.e., $\boldsymbol{\mu}_{\mathsf{d},L}(\mathbf{a}_{(i)}) = \boldsymbol{\mu}_{\mathsf{d},L}(i), i \in [L]$. In this way, $\boldsymbol{\mu}_{\mathsf{d},L}$ and $\mathcal{C}_L$ essentially defines a finite linear combination of Dirac measures. We then define a reduced problem of Problem PMO as follows:

$$\max_{\boldsymbol{\mu}_{\mathsf{d},L} \in \mathcal{U}_{\mathsf{d},L}, \mathcal{C}_L \in \mathcal{A}^L} \sum_{i=1}^L Q_\alpha^\star\left(\mathbf{s}, \mathbf{a}_{(i)}\right) \boldsymbol{\mu}_{\mathsf{d},L}(i)$$

$$(\widetilde{\mathsf{B}}_\alpha(\mathbf{s}, L))$$

$$\text{s.t.} \sum_{i=1}^L \mathbb{P}^\star(\mathbf{s}, \mathbf{a}_{(i)})\boldsymbol{\mu}_{\mathsf{d},L}(i) \geq 1 - \alpha, \ \mathbf{a}_{(i)} \in \mathcal{C}_L, \ \forall i \in [L].$$

Define $\widetilde{\mathcal{U}}_\alpha(\mathbf{s}, L) := \left\{(\boldsymbol{\mu}_{\mathsf{d},L}, \mathcal{C}_L) : \sum_{i=1}^L \mathbb{P}^\star(\mathbf{s}, \mathbf{a}_{(i)})\boldsymbol{\mu}_{\mathsf{d},L}(i) \geq 1 - \alpha\right\}$ as the feasible set of Problem $\widetilde{\mathsf{B}}_\alpha(\mathbf{s}, L)$. Since the constraint function $\sum_{i=1}^L \mathbb{P}^\star(\mathbf{s}, \mathbf{a}_{(i)})\boldsymbol{\mu}_{\mathsf{d},L}(i) : \mathcal{U}_{\mathsf{d},L} \times \mathcal{A}^L \to \mathbb{R}$ is continuous, and its domain $\mathcal{U}_{\mathsf{d},L} \times \mathcal{A}^L$ is compact, we have the feasible set $\widetilde{\mathcal{U}}_\alpha(\mathbf{s}, L)$ of Problem $\widetilde{\mathsf{B}}_\alpha(\mathbf{s}, L)$ is also a compact set. As a result, Problem $\widetilde{\mathsf{B}}_\alpha(\mathbf{s}, L)$'s optimal solution exists. We have the following proposition for the relationship between Problems PMO and $\widetilde{\mathsf{B}}_\alpha(\mathbf{s}, L)$:

**Proposition 1.** *Suppose that Assumptions 1 and 2 hold. Then, there exists a finite and positive integer $L < \infty$ such that makes each optimal solution of Problem $\widetilde{\mathsf{B}}_\alpha(\mathbf{s}, L)$ be an optimal solution of Problem* PMO.

*Proof of Proposition 1.* By Assumption 1, We know that $\mathcal{A}$ is compact and $Q^\star(\mathbf{s}, \mathbf{a})$ is continuous on $\mathcal{S} \times \mathcal{A}$ (from the one that $r(\mathbf{s}, \mathbf{a})$ is continuous on $\mathcal{S} \times \mathcal{A}$). Besides, by Assumption 2, we know that $\mathbb{P}^\star(\mathbf{s}, \mathbf{a})$ is continuous on $\mathcal{S} \times \mathcal{A}$ [25]. Then, the conclusion of Proposition 1 can be directly obtained by applying Theorem 1.3 of [26]. $\qquad\square$

For $L = 1$, Problem $\widetilde{\mathsf{B}}_\alpha(\mathbf{s}, L)$ becomes a chance-constrained optimization problem in $\mathcal{A}$:

$$\max_{\mathbf{a} \in \mathcal{A}} Q_\alpha^\star(\mathbf{s}, \mathbf{a}) \tag{$\mathsf{C}_\alpha(\mathbf{s})$}$$
$$\text{s.t. } \mathbb{P}^\star(\mathbf{s}, \mathbf{a}) \geq 1 - \alpha.$$

Let $J_\alpha^\star(\mathbf{s})$ be the optimal solution of Problem $\mathsf{C}_\alpha(\mathbf{s})$. For a given number $L \in \mathbb{N}_+$, let $\mathcal{E}_L := \left(\tilde{\alpha}^{(1)}, ..., \tilde{\alpha}^{(L)}\right)$ be an element of $[0, 1]^L$, defining as a set of violation probabilities, where each $\tilde{\alpha}^{(i)}$ is a threshold of violation probability in Problem $\mathsf{C}_\alpha(\mathbf{s})$ when $\alpha = \tilde{\alpha}^{(i)}$. For a violation probability set $\mathcal{E}_L$, we have a corresponding optimal objective value set $\{J_{\tilde{\alpha}^{(1)}}^\star(\mathbf{s}), ..., J_{\tilde{\alpha}^{(i)}}^\star(\mathbf{s}), ..., J_{\tilde{\alpha}^{(L)}}^\star(\mathbf{s})\}$, where $J_{\tilde{\alpha}^{(i)}}^\star(\mathbf{s})$ is the optimal objective value of Problem $\mathsf{C}_\alpha(\mathbf{s})$ when $\alpha = \tilde{\alpha}^{(i)}$. Let $\mathcal{V}_{\mathsf{d}, L} := \{\boldsymbol{\nu}_{\mathsf{d}, L} \in [0, 1]^L : \sum_{i=1}^L \boldsymbol{\nu}_{\mathsf{d}, L}(i) = 1\}$ be a set of discrete probability measures that defined on $\mathcal{E}_L$. By determining a violation probability set $\mathcal{E}_L$ and assigning a discrete probability $\boldsymbol{\nu}_{\mathsf{d}, L}$ to $\mathcal{E}_L$, we get a probabilistic decision in which the threshold of violation probability is randomly extracted from $\mathcal{E}_L$ obeying the discrete probability $\boldsymbol{\nu}_{\mathsf{d}, L}$. The corresponding expectation of the optimal objective value is $\sum_{i=1}^L J_{\tilde{\alpha}^{(i)}}^\star \boldsymbol{\nu}_{\mathsf{d}, L}(i)$. Another discrete probability measure optimization problem with chance constraint is formulated as

$$\max_{\boldsymbol{\nu}_{\mathsf{d}, L} \in \mathcal{V}_{\mathsf{d}, L}, \mathcal{E}_L \in [0, 1]^L} \sum_{i=1}^L J_{\tilde{\alpha}^{(i)}}^\star(\mathbf{s}) \boldsymbol{\nu}_{\mathsf{d}, L}(i) \tag{$\widetilde{\mathsf{V}}_\alpha(\mathbf{s}, L)$}$$
$$\text{s.t. } \sum_{i=1}^L (1 - \tilde{\alpha}^{(i)}) \boldsymbol{\nu}_{\mathsf{d}, L}(i) \geq 1 - \alpha, \ \tilde{\alpha}^{(i)} \in \mathcal{E}_L, \ \forall i \in [L].$$

We have the following proposition regarding the optimal values of Problems $\widetilde{\mathsf{B}}_\alpha(\mathbf{s}, L)$ and $\widetilde{\mathsf{V}}_\alpha(\mathbf{s}, L)$.

**Proposition 2.** *For every $L \in \mathbb{N}_+$ and $\mathbf{s} \in \mathcal{S}$, the optimal value of Problem $\widetilde{\mathsf{V}}_\alpha(\mathbf{s}, L)$ is equal to the one of Problem $\widetilde{\mathsf{B}}_\alpha(\mathbf{s}, L)$.*

*Proof of Proposition 2.* For an arbitrary $L \in \mathbb{N}_+$ and an arbitrary $\mathbf{s} \in \mathcal{S}$, let $\left(\tilde{\boldsymbol{\mu}}_{\mathsf{d}, L}, \tilde{\mathcal{C}}_L\right)$ be an optimal solution of Problem $\widetilde{\mathsf{B}}_\alpha(\mathbf{s}, L)$, where $\tilde{\mathcal{C}}_L = \left(\mathbf{a}_{(1)}, ..., \mathbf{a}_{(i)}, ..., \mathbf{a}_{(L)}\right)$. Notice that $\left(\tilde{\boldsymbol{\mu}}_{\mathsf{d}, L}, \tilde{\mathcal{C}}_L\right)$ is feasible for Problem $\widetilde{\mathsf{B}}_\alpha(\mathbf{s}, L)$ and thus we have

$$\sum_{i=1}^L \mathbb{P}^\star(\mathbf{s}, \mathbf{a}_{(i)}) \tilde{\boldsymbol{\mu}}_{\mathsf{d}, L}(i) \geq 1 - \alpha. \tag{8}$$

Define the optimal value of Problem $\widetilde{\mathsf{B}}_\alpha(\mathbf{s}, L)$ by $\widetilde{\mathcal{J}}_\alpha(\mathbf{s}, L)$ and thus

$$\widetilde{\mathcal{J}}_\alpha^{\mathsf{b}}(\mathbf{s}, L) = \sum_{i=1}^L Q_\alpha^\star\left(\mathbf{s}, \mathbf{a}_{(i)}\right) \tilde{\boldsymbol{\mu}}_{\mathsf{d}, L}(i). \tag{9}$$

Define a set of violation probabilities as

$$\mathcal{E}_L = \left(\tilde{\alpha}^{(1)}, ..., \tilde{\alpha}^{(L)}\right), \tag{10}$$

where $\tilde{\alpha}^{(i)} = 1 - \mathbb{P}^\star(\mathbf{s}, \mathbf{a}_{(i)})$. Let $\boldsymbol{\nu}_{\mathsf{d}, L} \in \mathcal{V}_{\mathsf{d}, L}$ be a probability measure that satisfies $\boldsymbol{\nu}_{\mathsf{d}, L}(i) = \tilde{\boldsymbol{\mu}}_{\mathsf{d}, L}(i), i \in [L]$. Then, by replacing $\tilde{\alpha}^{(i)} = 1 - \mathbb{P}^\star(\mathbf{s}, \mathbf{a}_{(i)})$ and $\boldsymbol{\nu}_{\mathsf{d}, L}(i) = \tilde{\boldsymbol{\mu}}_{\mathsf{d}, L}(i)$ into (8), we have

$$\sum_{i=1}^L (1 - \tilde{\alpha}^{(i)}) \boldsymbol{\nu}_{\mathsf{d}, L}(i) \geq 1 - \alpha, \tag{11}$$

which implies that $(\boldsymbol{\nu}_{\mathsf{d}, L}, \mathcal{E}_L)$ is a feasible solution of Problem $\widetilde{\mathsf{V}}_\alpha(\mathbf{s}, L)$. Let $\widetilde{\mathcal{J}}_\alpha^{\mathsf{v}}(\mathbf{s}, L)$ be the optimal value of Problem $\widetilde{\mathsf{V}}_\alpha(\mathbf{s}, L)$. Then, we have

$$\widetilde{\mathcal{J}}_\alpha^{\mathsf{v}}(\mathbf{s}, L) \geq \sum_{i=1}^L J_{\tilde{\alpha}^{(i)}}^\star(\mathbf{s}) \boldsymbol{\nu}_{\mathsf{d}, L}(i)$$
$$\geq \sum_{i=1}^L Q_\alpha^\star\left(\mathbf{s}, \mathbf{a}_{(i)}\right) \tilde{\boldsymbol{\mu}}_{\mathsf{d}, L}(i) \qquad \left(\text{From } J_{\tilde{\alpha}^{(i)}}^\star(\mathbf{s}) \geq Q_\alpha^\star\left(\mathbf{s}, \mathbf{a}_{(i)}\right), \forall \mathbf{s}\right)$$
$$= \widetilde{\mathcal{J}}_\alpha^{\mathsf{b}}(\mathbf{s}, L). \tag{12}$$

On the other hand, for an arbitrary $L \in \mathbb{N}_+$ and an arbitrary $\mathbf{s} \in \mathcal{S}$, let $\left( \bar{\boldsymbol{\nu}}_{\mathsf{d},L}, \overline{\mathcal{E}}_L \right)$ be an optimal solution of Problem $\widetilde{\mathsf{V}}_\alpha(\mathbf{s}, L)$, where $\overline{\mathcal{E}}_L = \left( \bar{\alpha}^{(1)}, ..., \bar{\alpha}^{(i)}, ..., \bar{\alpha}^{(L)} \right)$. We have

$$\widetilde{\mathcal{J}}_\alpha^{\mathsf{v}}(\mathbf{s}, L) = \sum_{i=1}^{L} J_{\bar{\alpha}^{(i)}}^{\star}(\mathbf{s}) \bar{\nu}_{\mathsf{d},L}(i), \tag{13}$$

$$\sum_{i=1}^{L} (1 - \bar{\alpha}^{(i)}) \bar{\nu}_{\mathsf{d},L}(i) \geq 1 - \alpha. \tag{14}$$

For $\overline{\mathcal{E}}_L$, define a set of decision variables as $\widehat{\mathcal{C}}_L = \{ \hat{\mathbf{a}}_{(1)}, ..., \hat{\mathbf{a}}_{(L)} \}$, where $\hat{\mathbf{a}}_{(i)}$ is an optimal solution of Problem $\mathsf{C}_\alpha(\mathbf{s})$ with $\alpha = \bar{\alpha}^{(i)}$. Note that we have $\mathbb{P}^{\star}(\mathbf{s}, \hat{\mathbf{a}}_{(i)}) \geq 1 - \bar{\alpha}^{(i)}$ and $Q^{\star}(\mathbf{s}, \hat{\mathbf{a}}_{(i)}) = J_{\bar{\alpha}^{(i)}}^{\star}(\mathbf{s})$. Define a discrete probability vector $\hat{\boldsymbol{\mu}}_{\mathsf{d},L} = \bar{\boldsymbol{\nu}}_{\mathsf{d},L}$. Since it holds that

$$\sum_{i=1}^{L} \mathbb{P}^{\star}(\mathbf{s}, \hat{\mathbf{a}}_{(i)}) \hat{\mu}_{\mathsf{d},L}(i) \geq \sum_{i=1}^{L} (1 - \bar{\alpha}^{(i)}) \bar{\nu}_{\mathsf{d},L}(i) \geq 1 - \alpha, \tag{15}$$

we have that $\left( \hat{\boldsymbol{\mu}}_{\mathsf{d},L}, \widehat{\mathcal{C}}_L \right)$ is a feasible solution of Problem $\widetilde{\mathsf{B}}_\alpha(\mathbf{s}, L)$. Therefore,

$$\begin{aligned} \widetilde{\mathcal{J}}_\alpha^{\mathsf{b}}(\mathbf{s}, L) &\geq \sum_{i=1}^{L} Q_\alpha^{\star} \left( \mathbf{s}, \hat{\mathbf{a}}_{(i)} \right) \hat{\mu}_{\mathsf{d},L}(i) \\ &= \sum_{i=1}^{L} J_{\bar{\alpha}^{(i)}}^{\star}(\mathbf{s}) \bar{\nu}_{\mathsf{d},L}(i) \qquad \left( \text{From } Q^{\star}(\mathbf{s}, \hat{\mathbf{a}}_{(i)}) = J_{\bar{\alpha}^{(i)}}^{\star}(\mathbf{s}), \ \hat{\boldsymbol{\mu}}_{\mathsf{d},L} = \bar{\boldsymbol{\nu}}_{\mathsf{d},L} \right) \\ &= \widetilde{\mathcal{J}}_\alpha^{\mathsf{v}}(\mathbf{s}, L). \end{aligned} \tag{16}$$

By (12) and (16), we have $\widetilde{\mathcal{J}}_\alpha^{\mathsf{b}}(\mathbf{s}, L) = \widetilde{\mathcal{J}}_\alpha^{\mathsf{v}}(\mathbf{s}, L)$, which completes the proof. $\qquad \square$

Based on Theorem 1, Propositions 1 and 2, we give the proof of Theorem 2 as follows:

*Proof of Theorem 2.* We first show that the optimal objective value $\widetilde{\mathcal{J}}_\alpha^{\mathsf{w}}(\mathbf{s})$ of Problem $\mathsf{W}_\alpha(\mathbf{s})$ satisfies

$$\widetilde{\mathcal{J}}_\alpha^{\mathsf{w}}(\mathbf{s}) = V_\alpha^{\star}(\mathbf{s}), \tag{17}$$

To attain (17), we will show

$$\widetilde{\mathcal{J}}_\alpha^{\mathsf{w}}(\mathbf{s}) \leq V_\alpha^{\star}(\mathbf{s}), \tag{18}$$

$$\widetilde{\mathcal{J}}_\alpha^{\mathsf{w}}(\mathbf{s}) \geq V_\alpha^{\star}(\mathbf{s}). \tag{19}$$

For (18), it can be directly obtained since $\widetilde{\mathcal{J}}_\alpha^{\mathsf{w}}(\mathbf{s}) = \widetilde{\mathcal{J}}_\alpha^{\mathsf{b}}(\mathbf{s}, L)$ with $L = 2$ and the feasible region of Problem $\widetilde{\mathsf{B}}_\alpha(\mathbf{s}, L)$ is a subset of the feasible region of Problem PMO with $L = 2$, which leads to $\widetilde{\mathcal{J}}_\alpha^{\mathsf{w}}(\mathbf{s}) = \widetilde{\mathcal{J}}_\alpha^{\mathsf{b}}(\mathbf{s}, L) \leq V_\alpha^{\star}(\mathbf{s})$. It remains to prove (19). We prove (19) in the following steps:

(a) We apply Proposition 1, supporting hyperplane theorem (p. 133 of [27]), and Caratheodory's theorem [17] to prove that $\widetilde{\mathcal{J}}_\alpha^{\mathsf{v}}(\mathbf{s}, L) \geq V_\alpha^{\star}(\mathbf{s})$ with $L = 2$;

(b) By Proposition 2, we obtain $\widetilde{\mathcal{J}}_\alpha^{\mathsf{w}}(\mathbf{s}) = \widetilde{\mathcal{J}}_\alpha^{\mathsf{b}}(\mathbf{s}, L) = \widetilde{\mathcal{J}}_\alpha^{\mathsf{v}}(\mathbf{s}, L) \geq V_\alpha^{\star}(\mathbf{s})$, which is (19).

Since (b) is obvious if (a) holds, we here focus on proving (a).

Define a set $\mathcal{H}(\mathbf{s}) := [0, 1] \times \mathbb{R}$. Let $(\tilde{\alpha}, -J_{\tilde{\alpha}}^{*}(\mathbf{s})) \in \mathcal{H}(\mathbf{s})$ be a pair of violation probability threshold $\tilde{\alpha}$ and the negative of the corresponding optimal value of Problem $\mathsf{C}_\alpha(\mathbf{s})$ with $\alpha = \tilde{\alpha}$. Let $\mathsf{conv} \left( \mathcal{H}(\mathbf{s}) \right)$ be the convex hull of $\mathcal{H}(\mathbf{s})$.

Construct a new optimization problem as

$$\min_{(\tilde{\alpha}_{\mathsf{h},\alpha}, -J_{\mathsf{h}}) \in \mathsf{conv}(\mathcal{H}(\mathbf{s}))} -J_{\mathsf{h}} \tag{$\mathsf{H}_\alpha(\mathbf{s})$}$$

$$\text{s.t. } \tilde{\alpha}_{\mathsf{h},\alpha} \leq \alpha.$$

Let $\left(\tilde{\alpha}^{\diamond}_{\mathsf{h},\alpha}(\mathbf{s}), -J^{\diamond}_{\mathsf{h},\alpha}(\mathbf{s})\right)$ be an optimal solution of Problem $\mathsf{H}_\alpha(\mathbf{s})$. We will show that $J^{\diamond}_{\mathsf{h},\alpha}(\mathbf{s}) \geq V^\star(\mathbf{s})$ for any $\mathbf{s} \in \mathcal{S}$, and $J^{\diamond}_{\mathsf{h},\alpha}(\mathbf{s}) \leq \widetilde{\mathcal{J}}^{\mathsf{v}}_\alpha(\mathbf{s}, L)$ with $L = 2$, which leads to (a).

First, we show that $J^{\diamond}_{\mathsf{h},\alpha}(\mathbf{s}) \geq V^\star(\mathbf{s})$ for any $\mathbf{s} \in \mathcal{S}$. For any $L \in \mathbb{N}_+$, let $\bar{\boldsymbol{\theta}}_L := \left(\bar{\boldsymbol{\nu}}_{\mathsf{d},L}, \bar{\alpha}^{(1)}, ..., \bar{\alpha}^{(L)}\right)$ be an optimal solution of Problem $\widetilde{\mathsf{V}}_\alpha(\mathbf{s}, L)$ and we have

$$\alpha_{\mathsf{mean}}(\bar{\boldsymbol{\theta}}_L) := \sum_{i=1}^{L} \bar{\alpha}^{(i)} \bar{\boldsymbol{\nu}}_{\mathsf{d},L}(i) \geq 1 - \alpha, \tag{20}$$

$$-J^{\star}_{\mathsf{mean}}(\bar{\boldsymbol{\theta}}_L) := \sum_{i=1}^{L} \left(-J^{\star}_{\bar{\alpha}^{(i)}}\right) \bar{\boldsymbol{\nu}}_{\mathsf{d},L}(i) = -\widetilde{\mathcal{J}}^{\mathsf{v}}_\alpha(\mathbf{s}, L). \tag{21}$$

By the definition of convex hull, we know that

$$\left(\alpha_{\mathsf{mean}}(\bar{\boldsymbol{\theta}}_S), -J^{*}_{\mathsf{mean}}(\bar{\boldsymbol{\theta}}_L)\right) \in \mathsf{conv}\left(\mathcal{H}(\mathbf{s})\right). \tag{22}$$

Due to (20), $\left(\alpha_{\mathsf{mean}}(\bar{\boldsymbol{\theta}}_L), -J^{\star}_{\mathsf{mean}}(\bar{\boldsymbol{\theta}}_L)\right)$ is a feasible solution of Problem $\mathsf{H}_\alpha(\mathbf{s})$ and thus we have

$$
\begin{aligned}
-J^{\diamond}_{\mathsf{h},\alpha}(\mathbf{s}) &\geq -J^{\star}_{\mathsf{mean}}(\bar{\boldsymbol{\theta}}_L) \\
&= -\widetilde{\mathcal{J}}^{\mathsf{v}}_\alpha(\mathbf{s}, L) \qquad\qquad \text{(By (21))} \\
&= -\widetilde{\mathcal{J}}^{\mathsf{b}}_\alpha(\mathbf{s}, L). \qquad\qquad \text{(By Proposition 2)}
\end{aligned} \tag{23}
$$

Note that (23) holds for an arbitrary $L \in \mathbb{N}_+$, which includes the one that satisfies $\widetilde{\mathcal{J}}^{\mathsf{b}}_\alpha(\mathbf{s}, L) = V^\star_\alpha(\mathbf{s})$ (by Proposition 1, there exists one $L$ that attains the equality). Therefore, we have

$$J^{\diamond}_{\mathsf{h},\alpha}(\mathbf{s}) \geq V^\star_\alpha(\mathbf{s}). \; \left(-J^{\diamond}_{\mathsf{h},\alpha}(\mathbf{s}) \leq -V^\star_\alpha(\mathbf{s})\right) \tag{24}$$

Then, we show that $J^{\diamond}_{\mathsf{h},\alpha}(\mathbf{s}) \leq \widetilde{\mathcal{J}}^{\mathsf{v}}_\alpha(\mathbf{s}, L)$ with $L = 2$. To attain this, we first show that $(\tilde{\alpha}^{\diamond}_{\mathsf{h},\alpha}(\mathbf{s}), -J^{\diamond}_{\mathsf{h},\alpha}(\mathbf{s}))$ is one boundary point of $\mathsf{conv}\left(\mathcal{H}(\mathbf{s})\right)$.

Suppose on the contrary that $(\tilde{\alpha}^{\diamond}_{\mathsf{h},\alpha}(\mathbf{s}), -J^{\diamond}_{\mathsf{h},\alpha}(\mathbf{s}))$ is an interior point. Thus there exists a neighborhood of $(\tilde{\alpha}^{\diamond}_{\mathsf{h},\alpha}(\mathbf{s}), -J^{\diamond}_{\mathsf{h},\alpha}(\mathbf{s}))$ such that is within $\mathsf{conv}(\mathcal{H}(\mathbf{s}))$. Suppose that $\mathcal{B}_\varepsilon\left((\tilde{\alpha}^{\diamond}_{\mathsf{h},\alpha}(\mathbf{s}), -J^{\diamond}_{\mathsf{h},\alpha}(\mathbf{s}))\right) \subset \mathsf{conv}(\mathcal{H}(\mathbf{s}))$, where $\varepsilon > 0$. For any $\tilde{\varepsilon} < \varepsilon$, we have that $(\tilde{\alpha}^{\diamond}_{\mathsf{h},\alpha}(\mathbf{s}), -J^{\diamond}_{\mathsf{h},\alpha}(\mathbf{s}) - \tilde{\varepsilon}) \in \mathcal{B}_\varepsilon\left((\tilde{\alpha}^{\diamond}_{\mathsf{h},\alpha}(\mathbf{s}), -J^{\diamond}_{\mathsf{h},\alpha}(\mathbf{s}))\right) \subset \mathsf{conv}(\mathcal{H}(\mathbf{s}))$. Since $1 - \tilde{\alpha}^{\diamond}_{\mathsf{h},\alpha}(\mathbf{s}) \geq 1 - \alpha$, $\left(\tilde{\alpha}^{\diamond}_{\mathsf{h},\alpha}(\mathbf{s}), -J^{\diamond}_{\mathsf{h},\alpha} - \tilde{\varepsilon}(\mathbf{s})\right)$ is a feasible solution of Problem $\mathsf{H}_\alpha(\mathbf{s})$. However, $-J^{\diamond}_{\mathsf{h},\alpha}(\mathbf{s}) - \tilde{\varepsilon} < -J^{\diamond}_{\mathsf{h},\alpha}(\mathbf{s})$ holds and it contracts with that $\left(\tilde{\alpha}^{\diamond}_{\mathsf{h},\alpha}(\mathbf{s}), -J^{\diamond}_{\mathsf{h},\alpha}(\mathbf{s})\right)$ is an optimal solution of Problem $\mathsf{H}_\alpha(\mathbf{s})$. Thus, $\left(\tilde{\alpha}^{\diamond}_{\mathsf{h},\alpha}(\mathbf{s}), -J^{\diamond}_{\mathsf{h},\alpha}(\mathbf{s})\right)$ is a boundary point.

By supporting hyperplane theorem (p. 133 of [27]), there exists a line $\mathcal{L}$ that passes through the boundary point $\left(\tilde{\alpha}^{\diamond}_{\mathsf{h},\alpha}(\mathbf{s}), -J^{\diamond}_{\mathsf{h}}(\mathbf{s})\right)$ and contains $\mathsf{conv}(\mathcal{H}(\mathbf{s}))$ in one of its closed half-spaces. Note that $\mathcal{L}$ is a one-dimensional linear space. Therefore, we can also say $\left(\tilde{\alpha}^{\diamond}_{\mathsf{h},\alpha}(\mathbf{s}), -J^{\diamond}_{\mathsf{h}}(\mathbf{s})\right)$ is within the convex hull of $\mathcal{L} \bigcap \mathcal{H}(\mathbf{s})$, $\mathsf{conv}(\mathcal{L} \bigcap \mathcal{H}(\mathbf{s}))$, namely, $\left(\tilde{\alpha}^{\diamond}_{\mathsf{h},\alpha}(\mathbf{s}), -J^{\diamond}_{\mathsf{h}}\right) \in \mathsf{conv}(\mathcal{L} \bigcap \mathcal{H}(\mathbf{s}))$. By Caratheodory's theorem [17], we have that $\left(\tilde{\alpha}^{\diamond}_{\mathsf{h},\alpha}(\mathbf{s}), -J^{\diamond}_{\mathsf{h}}(\mathbf{s})\right) \in \mathsf{conv}(\mathcal{L} \bigcap \mathcal{H}(\mathbf{s}))$ is within the convex combination of at most two points in $\mathcal{L} \bigcap \mathcal{H}$, namely, $\exists \boldsymbol{\nu}^{\diamond}_{\mathsf{m}} \in \mathcal{V}_{\mathsf{d},2}$, $\exists \{\tilde{\alpha}^{(1)}_{\mathsf{m}}, \tilde{\alpha}^{(2)}_{\mathsf{m}}\} \in [0,1]^2$ such that

$$-J^{\diamond}_{\mathsf{h},\alpha}(\mathbf{s}) = -J^{\star}_{\tilde{\alpha}^{(1)}_{\mathsf{m}}}(\mathbf{s})\boldsymbol{\nu}^{\diamond}_{\mathsf{m}}(1) - J^{\star}_{\tilde{\alpha}^{(2)}_{\mathsf{m}}}\boldsymbol{\nu}^{\diamond}_{\mathsf{m}}(2), \; \tilde{\alpha}^{\diamond}_{\mathsf{h},\alpha} = \tilde{\alpha}^{(1)}_{\mathsf{m}}\boldsymbol{\nu}^{\diamond}_{\mathsf{m}}(1) + \tilde{\alpha}^{(2)}_{\mathsf{m}}\boldsymbol{\nu}^{\diamond}_{\mathsf{m}}(2).$$

It holds that

$$1 - \left(\tilde{\alpha}^{(1)}_{\mathsf{m}}\boldsymbol{\nu}^{\diamond}_{\mathsf{m}}(1) + \tilde{\alpha}^{(2)}_{\mathsf{m}}\boldsymbol{\nu}^{\diamond}_{\mathsf{m}}(2)\right) = 1 - \tilde{\alpha}^{\diamond}_{\mathsf{h},\alpha} \geq 1 - \alpha.$$

Thus, $\left(\boldsymbol{\nu}_{\mathsf{m}}^{\diamond}(1), \boldsymbol{\nu}_{\mathsf{m}}^{\diamond}(2), \tilde{\alpha}_{\mathsf{m}}^{(1)}, \tilde{\alpha}_{\mathsf{m}}^{(2)}\right)$ is a feasible solution of Problem $\widetilde{\mathsf{V}}_{\alpha}(\mathbf{s}, L)$ when $L = 2$. Therefore, we have

$$-J_{\mathsf{h},\alpha}^{\diamond}(\mathbf{s}) = \sum_{i=1}^{2} \left(-J_{\tilde{\alpha}_{\mathsf{m}}^{(1)}}^{\star}(\mathbf{s})\right) \boldsymbol{\nu}_{\mathsf{m}}^{\diamond}(i) \leq -\widetilde{\mathcal{J}}_{\alpha}^{\mathsf{v}}(\mathbf{s}, L), \ L = 2. \tag{25}$$

From $J_{\mathsf{h},\alpha}^{\diamond}(\mathbf{s}) \leq \widetilde{\mathcal{J}}_{\alpha}^{\mathsf{v}}(\mathbf{s}, L)$ and $J_{\mathsf{h},\alpha}^{\diamond}(\mathbf{s}) \geq V_{\alpha}^{\star}(\mathbf{s})$ (by (24)), we have $\widetilde{\mathcal{J}}_{\alpha}^{\mathsf{v}}(\mathbf{s}, L) \geq V_{\alpha}^{\star}(\mathbf{s})$. Since $\widetilde{\mathcal{J}}_{\alpha}^{\mathsf{b}}(\mathbf{s}, L) = \widetilde{\mathcal{J}}_{\alpha}^{\mathsf{v}}(\mathbf{s}, L)$ by Proposition 2, we have $\widetilde{\mathcal{J}}_{\alpha}^{\mathsf{b}}(\mathbf{s}, L) \geq V_{\alpha}^{\star}(\mathbf{s})$, which leads to $\widetilde{\mathcal{J}}_{\alpha}^{\mathsf{b}}(\mathbf{s}, L) = V_{\alpha}^{\star}(\mathbf{s})$ since $\widetilde{\mathcal{J}}_{\alpha}^{\mathsf{b}}(\mathbf{s}, L) \leq V_{\alpha}^{\star}(\mathbf{s})$ also holds. Note that $\widetilde{\mathcal{J}}_{\alpha}^{\mathsf{w}}(\mathbf{s}, L) = \widetilde{\mathcal{J}}_{\alpha}^{\mathsf{b}}(\mathbf{s}, L)$, we have (17). The rest of Theorem 2 can be shown by applying Theorem 1, which completes the proof of Theorem 2. $\square$

# D  Proof of Theorem 3

As preparation for proving Theorem 3, we first give the following proposition on the optimal solution of Problem FPO.

**Proposition 3.** *Let* $z_{\alpha}^{\star}(\mathbf{s}) = \left(\mathbf{a}_{(1)}^{\star}(\mathbf{s}), \mathbf{a}_{(2)}^{\star}(\mathbf{s}), w^{\star}(\mathbf{s})\right)$ *be an optimal solution of Problem* FPO. *Let* $\tilde{\alpha}^{(1)} = 1 - \mathbb{P}^{\star}(\mathbf{a}_{(1)}^{\star}(\mathbf{s}))$ *and* $\tilde{\alpha}^{(2)} = 1 - \mathbb{P}^{\star}(\mathbf{a}_{(2)}^{\star}(\mathbf{s}))$. *We have*

$$V_{\alpha}^{\star}(\mathbf{s}) = w^{\star}(\mathbf{s})\widetilde{V}_{\tilde{\alpha}^{(1)}}^{\mathsf{d}}(\mathbf{s}) + \left(1 - w^{\star}(\mathbf{s})\right)\widetilde{V}_{\tilde{\alpha}^{(2)}}^{\mathsf{d}}(\mathbf{s}). \tag{26}$$

*Proof of Proposition 3.* By repeating the proof of Theorem 1, we can show that $\widetilde{V}_{\alpha}^{\mathsf{d}}(\mathbf{s})$ equals to the optimal value of the following problem:

$$\max_{\mathbf{a} \in \mathcal{A}} \quad Q_{\alpha}^{\star}(\mathbf{s}, \mathbf{a}) \quad \text{s.t.} \quad \mathbb{P}^{\star}(\mathbf{s}, \mathbf{a}) \geq 1 - \alpha. \tag{27}$$

By Theorem 2, we have

$$V_{\alpha}^{\star}(\mathbf{s}) = w^{\star}(\mathbf{s})Q_{\alpha}^{\star}\left(\mathbf{s}, \mathbf{a}_{(1)}^{\star}(\mathbf{s})\right) + (1 - w^{\star}(\mathbf{s}))Q_{\alpha}^{\star}\left(\mathbf{s}, \mathbf{a}_{(2)}^{\star}(\mathbf{s})\right). \tag{28}$$

$\square$

Based on Proposition 3, we give the proof of Theorem 3.

*Proof of Theorem 3.* If $\alpha = 0$, the optimal solution of Problem $\widetilde{\mathsf{V}}_{\alpha}(\mathbf{s}, L)$ with $L = 2$ has to set $\tilde{\alpha}^{(1)} = \tilde{\alpha}^{(2)} = 0$, which leads to

$$\widetilde{J}_{0}^{\mathsf{v}}(\mathbf{s}) = J_{0}^{\star}(\mathbf{s}). \tag{29}$$

Then, by Proposition 2 and Theorem 2, we have

$$V_{0}^{\star}(\mathbf{s}) = \widetilde{J}_{0}^{\mathsf{w}}(\mathbf{s}) = \widetilde{J}_{0}^{\mathsf{b}}(\mathbf{s}) = \widetilde{J}_{0}^{\mathsf{v}}(\mathbf{s}) = J_{0}^{\star}(\mathbf{s}), \tag{30}$$

which can be attained by an optimal solution of Problem $\mathsf{C}_{\alpha}(\mathbf{s})$ referring to a deterministic policy. $\square$

# E  Proof of Theorem 4

*Proof.* Define the discounted return of a specific trajectory with $\gamma_{\mathsf{unsafe}} \in (0, 1)$ by

$$G(\boldsymbol{\tau}_{\infty}) := \sum_{i=1}^{\infty} \gamma_{\mathsf{unsafe}}^{i} \mathbb{I}(g(\mathbf{s}_{i})). \tag{31}$$

Define a function $\mathbb{G}^{\star}(\mathbf{s}, \mathbf{a})$ by

$$\mathbb{G}^{\star}(\mathbf{s}, \mathbf{a}) = \mathbb{E}_{\boldsymbol{\tau}_{\infty} \sim \mathrm{Pr}_{\mathbf{s}_0, \infty}^{\pi_{\mathsf{sec}}^{\star}}} \left\{G(\boldsymbol{\tau}_{\infty}) | \mathbf{s}_0 = \mathbf{s}, \mathbf{a}_0 = \mathbf{a}\right\}. \tag{32}$$

Here, $\boldsymbol{\pi}_{\text{sec}}^\star$ is an optimal solution of Problem ECRL. From the definition of $G(\boldsymbol{\tau}_\infty)$ in (31), we can rewrite $\mathbb{G}^\star(\mathbf{s}, \mathbf{a})$ in the following way:

$$
\begin{aligned}
\mathbb{G}^\star(\mathbf{s}, \mathbf{a}) &= \sum_{i=1}^\infty \gamma_{\text{unsafe}}^i \mathbb{E}_{\boldsymbol{\tau}_\infty \sim \text{Pr}_{\mathbf{s}_0, \infty}^{\boldsymbol{\pi}_{\text{sec}}^\star}} \left\{ \mathbb{I}(g(\mathbf{s}_i)) | \mathbf{s}_0 = \mathbf{s}, \mathbf{a}_0 = \mathbf{a} \right\} \\
&= \sum_{i=1}^\infty \gamma_{\text{unsafe}}^i \text{Pr}_{\mathbf{s}_0, \infty}^{\boldsymbol{\pi}_{\text{sec}}^\star} \left\{ \mathbb{I}(g(\mathbf{s}_i)) | \mathbf{s}_0 = \mathbf{s}, \mathbf{a}_0 = \mathbf{a} \right\} .
\end{aligned}
\tag{33}
$$

The continuity of each $\text{Pr}_{\mathbf{s}_0, \infty}^{\boldsymbol{\pi}_{\text{sec}}^\star} \left\{ \mathbb{I}(g(\mathbf{s}_i)) | \mathbf{s}_0 = \mathbf{s}, \mathbf{a}_0 = \mathbf{a} \right\}$ is guaranteed by Assumption 2 and the continuity of $g(\cdot)$ (pp. 78-79 of [25]), which naturally leads to the continuity of $\mathbb{G}^\star(\mathbf{s}, \mathbf{a})$. With $\mathbb{G}^\star(\mathbf{s}, \mathbf{a})$, a probability measure optimization problem is defined as follows:

$$
\begin{aligned}
\max_{\boldsymbol{\mu} \in M(\mathcal{A})} \quad & \int_{\mathcal{A}} Q_\alpha^\star(\mathbf{s}, \mathbf{a}) \, \mathsf{d}\boldsymbol{\mu} \\
\text{s.t.} \quad & \int_{\mathcal{A}} \mathbb{G}^\star(\mathbf{s}, \mathbf{a}) \, \mathsf{d}\boldsymbol{\mu} \leq \alpha.
\end{aligned}
\tag{$\text{B}_\alpha^{\text{sec}}(\mathbf{s})$}
$$

By just repeating the proof of Theorem 1, we can obtain that the optimal objective value of Problem $\text{B}_\alpha^{\text{sec}}(\mathbf{s})$ equals the one of Problem ECRL for any $\mathbf{s} \in \mathcal{S}$. A flipping-based version of Problem $\text{B}_\alpha^{\text{sec}}(\mathbf{s})$ is written by

$$
\begin{aligned}
\max_{\mathbf{a}_{(1)}, \mathbf{a}_{(2)}, w} \quad & w Q_\alpha^\star(\mathbf{s}, \mathbf{a}_{(1)}) + (1-w) Q_\alpha^\star(\mathbf{s}, \mathbf{a}_{(2)}) \\
\text{s.t.} \quad & w \mathbb{G}^\star(\mathbf{s}, \mathbf{a}_{(1)}) + (1-w) \mathbb{G}^\star(\mathbf{s}, \mathbf{a}_{(2)}) \leq \alpha.
\end{aligned}
\tag{$\text{W}_\alpha^{\text{sec}}(\mathbf{s})$}
$$

The continuity of $\mathbb{G}^\star(\mathbf{s}, \mathbf{a})$ holds and it is bounded within $[0, 1]$. Thus, Theorem 4 can be proved by following the same process of proving Theorem 2 after replacing $\mathbb{P}^\star(\mathbf{s}, \mathbf{a})$ by $\mathbb{G}^\star(\mathbf{s}, \mathbf{a})$. $\qquad \square$

# F  Proof of Theorem 5

*Proof.* Define a violation probability function $\mathbb{V}_{\text{joint}}(\boldsymbol{\theta}, \mathbf{s})$ by

$$
\mathbb{V}_{\text{joint}}(\boldsymbol{\theta}, \mathbf{s}) = \text{Pr}_{\mathbf{s}, \infty}^{\boldsymbol{\pi}_{\boldsymbol{\theta}}} \left\{ \mathbf{s}_i \notin \mathbb{S}, \exists i \in [T] \, | \, \mathbf{s}_0 = \mathbf{s} \in \mathbb{S} \right\}.
\tag{34}
$$

Note that the constraint $\mathbb{V}_{\text{joint}}(\boldsymbol{\theta}, \mathbf{s}) \leq \alpha$ is equivalent to

$$
\text{Pr}_{\mathbf{s}, \infty}^{\boldsymbol{\pi}_{\boldsymbol{\theta}}} \left\{ \mathbf{s}_i \in \mathbb{S}, \forall i \in [T] \, | \, \mathbf{s}_0 = \mathbf{s} \in \mathbb{S} \right\} \geq 1 - \alpha.
$$

By using Boole's inequality, we have

$$
\begin{aligned}
\mathbb{V}_{\text{joint}}(\boldsymbol{\theta}, \mathbf{s}) &\leq \sum_{i=1}^T \text{Pr}_{\mathbf{s}, \infty}^{\boldsymbol{\pi}_{\boldsymbol{\theta}}} \left\{ \mathbf{s}_i \notin \mathbb{S} \, | \, \mathbf{s}_0 \in \mathbb{S} \right\} \\
&= \sum_{i=1}^T \mathbb{E}_{\boldsymbol{\tau}_\infty \sim \text{Pr}_{\mathbf{s}, \infty}^{\boldsymbol{\pi}_{\boldsymbol{\theta}}}} \left\{ \mathbb{I}(g((\mathbf{s}_i))) \right\} \\
&= \mathbb{E}_{\boldsymbol{\tau}_\infty \sim \text{Pr}_{\mathbf{s}, \infty}^{\boldsymbol{\pi}_{\boldsymbol{\theta}}}} \left\{ \sum_{i=1}^T \mathbb{I}(g((\mathbf{s}_i))) \right\}.
\end{aligned}
\tag{35}
$$

Define $\widetilde{V}_{\text{unsafe}}^T$ by

$$
\widetilde{V}_{\text{unsafe}}^T(\boldsymbol{\theta}, \mathbf{s}) := \mathbb{E}_{\boldsymbol{\tau}_\infty \sim \text{Pr}_{\mathbf{s}, \infty}^{\boldsymbol{\pi}_{\boldsymbol{\theta}}}} \left\{ \sum_{i=1}^T \mathbb{I}(g((\mathbf{s}_i))) \right\}.
\tag{36}
$$

Due to (35), $\widetilde{V}_{\text{unsafe}}^T(\boldsymbol{\theta}, \mathbf{s}) \leq \alpha$, $\forall \mathbf{s} \in \mathbb{S}$ implies $\mathbb{V}_{\text{joint}}(\boldsymbol{\theta}, \mathbf{s}) \leq \alpha$, $\forall \mathbf{s} \in \mathbb{S}$. By replacing $\mathbf{s}$ by $\mathbf{s}_k$, we obtain (4) by setting $C(\boldsymbol{\theta}, \mathbf{s}) = \widetilde{V}_{\text{unsafe}}^T(\boldsymbol{\theta}, \mathbf{s}) - \alpha$. Then, $\widetilde{V}_{\text{unsafe}}^T(\boldsymbol{\theta}, \mathbf{s}) - \alpha$ is a conservative approximation of joint chance constraint (3).

Then, we will show that we can find $\gamma_{\text{unsafe}}$ and $T$ to make the following equality holds:

$$
H_{\text{unsafe}}(\boldsymbol{\theta}, \mathbf{s}) \geq \widetilde{V}_{\text{unsafe}}^T(\boldsymbol{\theta}, \mathbf{s}), \quad \forall \boldsymbol{\theta} \in \Theta, \, \mathbf{s} \in \mathbb{S}.
\tag{37}
$$

Define $\overline{V}_{\text{unsafe}}^{T, \gamma_{\text{unsafe}}}(\boldsymbol{\theta}, \mathbf{s})$ by

$$\overline{V}_{\text{unsafe}}^{T, \gamma_{\text{unsafe}}}(\boldsymbol{\theta}, \mathbf{s}) := \mathbb{E}_{\boldsymbol{\tau}_{\infty} \sim \text{Pr}_{\mathbf{s}, \infty}^{\pi_{\boldsymbol{\theta}}}} \left\{ \sum_{i=1}^{T} \gamma_{\text{unsafe}}^{i} \mathbb{I}\left(g((\mathbf{s}_i))\right) \right\}. \tag{38}$$

Define the error between $\overline{V}_{\text{unsafe}}^{T, \gamma_{\text{unsafe}}}(\boldsymbol{\theta}, \mathbf{s})$ and $H_{\text{unsafe}}(\boldsymbol{\theta}, \mathbf{s})$ by

$$\tilde{\epsilon}_V^{\text{inf}}(T, \gamma_{\text{unsafe}}, \boldsymbol{\theta}, \mathbf{s}) := H_{\text{unsafe}}(\boldsymbol{\theta}, \mathbf{s}) - \overline{V}_{\text{unsafe}}^{T, \gamma_{\text{unsafe}}}(\boldsymbol{\theta}, \mathbf{s}). \tag{39}$$

Note that $\tilde{\epsilon}_V^{\text{inf}}(T, \gamma_{\text{unsafe}}, \boldsymbol{\theta}, \mathbf{s})$ is positive for any $\boldsymbol{\theta} \in \Theta$, $\mathbf{s} \in \mathbb{S}, \gamma_{\text{unsafe}} \in (0, 1]$ and it decreases monotonically as $T$ increases. It increases monotonically as $\gamma_{\text{unsafe}}$ increases to 1.

On the other hand, define the error between $\overline{V}_{\text{unsafe}}^{T, \gamma_{\text{unsafe}}}(\boldsymbol{\theta}, \mathbf{s})$ and $\widetilde{V}_{\text{unsafe}}^{T}(\boldsymbol{\theta}, \mathbf{s})$ by

$$\tilde{\epsilon}_V^{T}(T, \gamma_{\text{unsafe}}, \boldsymbol{\theta}, \mathbf{s}) := \widetilde{V}_{\text{unsafe}}^{T}(\boldsymbol{\theta}, \mathbf{s}) - \overline{V}_{\text{unsafe}}^{T, \gamma_{\text{unsafe}}}(\boldsymbol{\theta}, \mathbf{s}). \tag{40}$$

The error $\tilde{\epsilon}_V^{T}(T, \gamma_{\text{unsafe}}, \boldsymbol{\theta}, \mathbf{s})$ decreases monotonically as $\gamma_{\text{unsafe}}$ increases to 1. It decreases monotonically as $T$ decreases.

We give the error between $H_{\text{unsafe}}(\boldsymbol{\theta}, \mathbf{s})$ and $\widetilde{V}_{\text{unsafe}}^{T}(\boldsymbol{\theta}, \mathbf{s})$ as follows:

$$\epsilon_V(T, \gamma_{\text{unsafe}}, \boldsymbol{\theta}, \mathbf{s}) := H_{\text{unsafe}}(\boldsymbol{\theta}, \mathbf{s}) - \widetilde{V}_{\text{unsafe}}^{T}(\boldsymbol{\theta}, \mathbf{s}) = \tilde{\epsilon}_V^{\text{inf}}(T, \gamma_{\text{unsafe}}, \boldsymbol{\theta}, \mathbf{s}) - \tilde{\epsilon}_V^{T}(T, \gamma_{\text{unsafe}}, \boldsymbol{\theta}, \mathbf{s}). \tag{41}$$

For any given $\boldsymbol{\theta}, \mathbf{s}$, it is able to decrease $T$ and meanwhile increase $\gamma_{\text{unsafe}}$ to simultaneously achieve:

- increasing $\tilde{\epsilon}_V^{\text{inf}}(T, \gamma_{\text{unsafe}}, \boldsymbol{\theta}, \mathbf{s})$;
- decreasing $\tilde{\epsilon}_V^{T}(T, \gamma_{\text{unsafe}}, \boldsymbol{\theta}, \mathbf{s})$.

Then, there is small enough $T$ and large enough $\gamma_{\text{unsafe}}$ to ensure that $\epsilon_V(T, \gamma_{\text{unsafe}}, \boldsymbol{\theta}, \mathbf{s}) > 0$. Besides, since $\Theta$ and $\mathbb{S}$ are compact and functions $H_{\text{unsafe}}(\boldsymbol{\theta}, \mathbf{s})$, $\widetilde{V}_{\text{unsafe}}^{T}(\boldsymbol{\theta}, \mathbf{s})$, and $\overline{V}_{\text{unsafe}}^{T, \gamma_{\text{unsafe}}}(\boldsymbol{\theta}, \mathbf{s})$ are continuous (yielded by Assumption 2), $\overline{T}$ and $\underline{\gamma}_{\text{unsafe}}$ such that, if $\gamma_{\text{unsafe}} > \underline{\gamma}_{\text{unsafe}}$ and $T < \overline{T}$, $\epsilon_V(T, \gamma_{\text{unsafe}}, \boldsymbol{\theta}, \mathbf{s}) > 0$, $\forall \boldsymbol{\theta} \in \Theta, \mathbf{s} \in \mathbb{S}$. Then, we have

$$H_{\text{unsafe}}(\boldsymbol{\theta}, \mathbf{s}) - \alpha \leq 0 \Rightarrow \widetilde{V}_{\text{unsafe}}^{T}(\boldsymbol{\theta}, \mathbf{s}) + \epsilon_V(T, \gamma_{\text{unsafe}}, \boldsymbol{\theta}, \mathbf{s}) - \alpha \leq 0 \Rightarrow \widetilde{V}_{\text{unsafe}}^{T}(\boldsymbol{\theta}, \mathbf{s}) - \alpha \leq 0.$$

Thus, by replacing $\mathbf{s}$ by $\mathbf{s}_k$, we obtain (3) by setting $C(\boldsymbol{\theta}, \mathbf{s}) = H_{\text{unsafe}}(\boldsymbol{\theta}, \mathbf{s}) - \alpha$ if $\gamma_{\text{unsafe}} > \underline{\gamma}_{\text{unsafe}}$ and $T < \overline{T}$. Then, $H_{\text{unsafe}}(\boldsymbol{\theta}, \mathbf{s}) - \alpha$ is a conservative approximation of joint chance constrain (3). $\square$

## G  Proof of Theorem 6

As a preparation for proving Theorem 6, we first give the following proposition for the optimal solution of Problem PFPRL.

**Proposition 4.** *Let* $\boldsymbol{\zeta}_{\text{m}}^{*} = \left(\nu_{\text{m}}^{*}(1), \nu_{\text{m}}^{*}(2), \boldsymbol{\theta}_{*}^{(1)}, \boldsymbol{\theta}_{*}^{(2)}\right) \in D_{\alpha}$ *be an optimal solution of Problem* PFPRL. *Let* $\tilde{\alpha}^{(1)} = 1 - F^{\text{d}}(\boldsymbol{\theta}_{*}^{(1)})$ *and* $\tilde{\alpha}^{(2)} = 1 - F^{\text{d}}(\boldsymbol{\theta}_{*}^{(2)})$. *We have* $\boldsymbol{\theta}_{*}^{(1)} \in \Theta_{\tilde{\alpha}^{(1)}}^{*}, \boldsymbol{\theta}_{*}^{(1)} \in \Theta_{\tilde{\alpha}^{(2)}}^{*}$.

*Proof of Proposition 4.* Suppose that $\boldsymbol{\theta}_{*}^{(1)} \notin \Theta_{\tilde{\alpha}^{(1)}}^{*}$. Then, $J(\boldsymbol{\theta}_{*}^{(1)}) < J_{\tilde{\alpha}^{(1)}}^{*}$ holds. Let $\hat{\boldsymbol{\theta}}^{(1)} \in \Theta_{\tilde{\alpha}^{(1)}}^{*}$. Then, $F^{\text{d}}(\hat{\boldsymbol{\theta}}^{(1)}) \geq 1 - \tilde{\alpha}^{(1)} = F^{\text{d}}(\boldsymbol{\theta}_{*}^{(1)})$ and $J(\hat{\boldsymbol{\theta}}^{(1)}) = J_{\tilde{\alpha}^{(1)}}^{*} > J(\boldsymbol{\theta}_{*}^{(1)})$. We have

$$F^{\text{d}}(\hat{\boldsymbol{\theta}}^{(1)})\nu_{\text{m}}^{*}(1) + F^{\text{d}}(\boldsymbol{\theta}_{*}^{(2)})\nu_{\text{m}}^{*}(2) \geq \sum_{i=1}^{2} F^{\text{d}}(\boldsymbol{\theta}_{*}^{(i)})\nu_{\text{m}}^{*}(i) \geq 1 - \alpha,$$

$$J(\hat{\boldsymbol{\theta}}^{(1)})\nu_{\text{m}}^{*}(1) + J(\boldsymbol{\theta}_{*}^{(2)})\nu_{\text{m}}^{*}(2) > \sum_{i=1}^{2} J(\boldsymbol{\theta}_{*}^{(i)})\nu_{\text{m}}^{*}(i).$$

Thus, $\hat{\boldsymbol{\zeta}}_{\text{m}} = \left(\nu_{\text{m}}^{*}(1), \nu_{\text{m}}^{*}(2), \hat{\boldsymbol{\theta}}^{(1)}, \boldsymbol{\theta}_{*}^{(2)}\right)$ is a feasible solution of Problem PFPRL and has a larger objective function value than $\boldsymbol{\zeta}_{\text{m}}^{*}$ which contradicts to $\boldsymbol{\zeta}_{\text{m}}^{*} \in D_{\alpha}$. Therefore, we have $\boldsymbol{\theta}_{*}^{(1)} \in \Theta_{\tilde{\alpha}^{(1)}}^{*}$. Follow the above procedures, we could also prove that $\boldsymbol{\theta}_{*}^{(2)} \in \Theta_{\tilde{\alpha}^{(2)}}^{*}$. $\square$

Then, we give the proof of Theorem 6 as follows:

*Proof of Theorem 6.* For $\mathcal{J}_\alpha^* = \mathcal{J}_\alpha^w$, it can be obtained by repeating the proof of Theorem 2.

Let $\zeta_m^* = \left( \nu_m^*(1), \nu_m^*(2), \boldsymbol{\theta}_*^{(1)}, \boldsymbol{\theta}_*^{(2)} \right) \in D_\alpha$ be an optimal solution of Problem PFPRL. Let $\tilde{\alpha}^{(1)} = 1 - F^d(\boldsymbol{\theta}_*^{(1)})$ and $\tilde{\alpha}^{(2)} = 1 - F^d(\boldsymbol{\theta}_*^{(2)})$. By Theorem 6 and Proposition 4, we have

$$\mathcal{J}_\alpha^* = \nu_m^*(1) * J_{\tilde{\alpha}^{(1)}}^* + \nu_m^*(2) * J_{\tilde{\alpha}^{(2)}}^*. \tag{42}$$

Since $J_\alpha^*$ is a strictly convex function of $\alpha$ on $(\underline{\alpha}, \overline{\alpha})$, we have

$$\begin{aligned} J_\alpha^* &< \nu_m^*(1) * J_{\hat{\alpha}^{(1)}}^* + \nu_m^*(2) * J_{\hat{\alpha}^{(2)}}^* \quad \left( \hat{\alpha}^{(1)}, \hat{\alpha}^{((2)} \in (\underline{\alpha}, \overline{\alpha}) \right) \\ &\leq \nu_m^*(1) * J_{\tilde{\alpha}^{(1)}}^* + \nu_m^*(2) * J_{\tilde{\alpha}^{(2)}}^* \\ &= \mathcal{J}_\alpha^*, \end{aligned}$$

which completes the proof. $\qquad\square$

# H   Proof of Theorem 7

*Proof of Theorem 7.* Since Problem LP is a special case of Problem PSPRL, Theorem 6 implies the existence of an optimal solution in $\tilde{D}_\alpha(\mathcal{Z}_S)$ with no more than two non-zero elements.

Since $\tilde{\boldsymbol{\theta}}_i$ is the optimal solution of Problem PDPRL with $\alpha = \tilde{\alpha}_i$, Problem LP has the same optimal value with the following optimization problem:

$$\max_{\nu_s(1),\dots,\nu_s(S) \in [0,1]^S} \sum_{i=1}^S J_{\tilde{\alpha}_i}^* \nu_s(i) \tag{$\widetilde{\mathcal{K}}_\alpha(\mathcal{Z}_S)$}$$
$$\text{s.t.} \sum_{i=1}^S \nu_s(i)\tilde{\alpha}_i \leq \alpha, \ \sum_{i=1}^S \nu_s(i) = 1.$$

Define another optimization problem as:

$$\max_{\nu_c \in M([0,1])} \int_{[0,1]} J_{\tilde{\alpha}}^* d\nu_c \tag{$\widehat{\mathcal{K}}_{\tilde{\alpha}}$}$$
$$\text{s.t.} \int_{[0,1]} \tilde{\alpha} d\nu_c \leq \alpha.$$

Let $\widehat{\mathcal{J}}_\alpha^k$ and $\widehat{D}_\alpha$ be the optimal solution and optimal solution set of Problem $\widehat{\mathcal{K}}_{\tilde{\alpha}}$.

Note that $\mathcal{Z}_S = \{\tilde{\alpha}_i\}_{i=1}^S$ is extracted according to uniform distribution. By applying Theorem 6 of [35], we have

- $\widetilde{\mathcal{J}}_\alpha^k(\mathcal{Z}_S) \leq \widehat{\mathcal{J}}_\alpha^k$;
- $\widetilde{\mathcal{J}}_\alpha^k(\mathcal{Z}_S) \to \widehat{\mathcal{J}}_\alpha^k$ with probability 1 as $S \to \infty$.

Then, if we show that $\widehat{\mathcal{J}}_\alpha^k = \mathcal{J}_\alpha^*$, it leads to $\widetilde{\mathcal{J}}_\alpha^k(\mathcal{Z}_S) \to \mathcal{J}_\alpha^*$ with probability 1 as $S \to \infty$.

By Proposition 4, we have

$$\mathcal{J}_\alpha^* = \sum_{i=1}^2 J(\boldsymbol{\theta}_*^{(i)})\nu_m^*(i) = \sum_{i=1}^2 J_{\tilde{\alpha}^{(i)}}^* \nu_m^*(i), \tag{43}$$

where $\boldsymbol{\theta}_*^{(i)}$ is one optimal solution of Problem PDPRL with $\alpha = \tilde{\alpha}^{(i)}, i = 1, 2$. We further have

$$\sum_{i=1}^2 \tilde{\alpha}^{(i)}\nu_m^*(i) \leq \alpha. \tag{44}$$

Thus, the probability measure $\tilde{\nu}_{\mathsf{c}}^{\mathsf{flip}}$ defined by

$$\tilde{\nu}_{\mathsf{c}}^{\mathsf{flip}}\left\{\tilde{\alpha}^{(1)}\right\} = \nu_{\mathsf{m}}^{*}(1), \ \tilde{\nu}_{\mathsf{c}}^{\mathsf{flip}}\left\{\tilde{\alpha}^{(2)}\right\} = \nu_{\mathsf{m}}^{*}(2) \tag{45}$$

is a feasible solution of Problem $\widehat{\mathcal{K}}_{\tilde{\alpha}}$, which implies that

$$\mathcal{J}_{\alpha}^{*} \leq \widehat{\mathcal{J}}_{\alpha}^{\mathsf{k}}. \tag{46}$$

On the other hand, by applying Theorem 6, we have that Problem $\widehat{\mathcal{K}}_{\tilde{\alpha}}$ has a flipping-based optimal solution $\hat{\nu}_{\mathsf{c}}^{\mathsf{flip}}$,

$$\hat{\nu}_{\mathsf{c}}^{\mathsf{flip}}\left\{\hat{\alpha}^{(1)}\right\} = \hat{\nu}_{\mathsf{m}}(1), \ \hat{\nu}_{\mathsf{c}}^{\mathsf{flip}}\left\{\hat{\alpha}^{(2)}\right\} = \hat{\nu}_{\mathsf{m}}(2) \tag{47}$$

It leads to

$$\widehat{\mathcal{J}}_{\alpha}^{\mathsf{k}} = \sum_{i=1}^{2} J_{\hat{\alpha}^{(i)}}^{*}\hat{\nu}_{\mathsf{m}}(i) = \sum_{i=1}^{2} J(\hat{\boldsymbol{\theta}}_{i})\hat{\nu}_{\mathsf{m}}(i) \tag{48}$$

$$\sum_{i=1}^{2} \hat{\nu}_{\mathsf{m}}(i)F^{\mathsf{d}}(\hat{\boldsymbol{\theta}}_{i}) \geq 1 - \alpha, \tag{49}$$

which implies that $\widehat{\mathcal{J}}_{\alpha}^{\mathsf{k}}$ equals an objective value of a feasible solution of Problem PSPRL. Thus,

$$\mathcal{J}_{\alpha}^{*} \geq \widehat{\mathcal{J}}_{\alpha}^{\mathsf{k}}. \tag{50}$$

Due to 46 and (50), we have $\widehat{\mathcal{J}}_{\alpha}^{\mathsf{k}} = \mathcal{J}_{\alpha}^{*}$, which completes the proof. $\qquad\square$

# I  Proof of Theorem 8

*Proof.* First, we show that $\tilde{\boldsymbol{\theta}}_{\alpha_{\mathsf{s}}}(\mathbf{s}, \boldsymbol{\theta}_{k}, \mathcal{D}_{N})$ is a feasible solution of Problem CPOS with probability larger than $1 - \exp\left\{-2N(\alpha_{\mathsf{s}} - \tilde{\alpha}_{\mathsf{s}})^{2}(1 - \gamma_{\mathsf{unsafe}})^{2}\right\}$. Define two functions by

$$F(\boldsymbol{\theta}) := 1 - \mathbb{E}_{\mathbf{s}_{\mathsf{ini}}\sim\boldsymbol{\pi}_{\boldsymbol{\theta}_{k}}, \mathbf{a}\sim\boldsymbol{\pi}_{\hat{\boldsymbol{\theta}}_{\mathsf{bad}}}}\left\{\mathbb{I}(g(\mathbf{s}^{+}))\right\},$$

$$\widetilde{F}(\boldsymbol{\theta}, \mathcal{D}_{N}) := 1 - \frac{1}{N}\sum_{i=1}^{N}\mathbb{I}(g(\mathbf{s}^{+,(i)})).$$

Here, we simplify the notation by omitting $\mathbf{s}$ and $\boldsymbol{\theta}_{k}$ since it claims the same conclusion for all $\mathbf{s}$ and $\boldsymbol{\theta}_{k}$. Besides, define two probability $\alpha_{\mathsf{s}}^{\mathsf{trf}}$ and $\tilde{\alpha}_{\mathsf{s}}^{\mathsf{trf}}$ transformed from $\alpha_{\mathsf{s}}$ and $\tilde{\alpha}_{\mathsf{s}}$ by

$$\alpha_{\mathsf{s}}^{\mathsf{trf}} := (\alpha_{\mathsf{s}} - H_{\mathsf{unsafe}}(\boldsymbol{\theta}_{k}, \mathbf{s}))(1 - \gamma_{\mathsf{unsafe}}).$$

$$\tilde{\alpha}_{\mathsf{s}}^{\mathsf{trf}} := (\tilde{\alpha}_{\mathsf{s}} - H_{\mathsf{unsafe}}(\boldsymbol{\theta}_{k}, \mathbf{s}))(1 - \gamma_{\mathsf{unsafe}}).$$

Note that $\alpha_{\mathsf{s}}^{\mathsf{trf}} - \tilde{\alpha}_{\mathsf{s}}^{\mathsf{trf}} = (\alpha_{\mathsf{s}} - \tilde{\alpha}_{\mathsf{s}})(1 - \gamma_{\mathsf{unsafe}})$.

Let $\hat{\boldsymbol{\theta}}_{\mathsf{bad}}$ be an infeasible solution of Problem CPOS. Then, we have

$$F(\hat{\boldsymbol{\theta}}_{\mathsf{bad}}) < 1 - \alpha_{\mathsf{s}}^{\mathsf{trf}}. \tag{51}$$

The probability of $\hat{\boldsymbol{\theta}}_{\mathsf{bad}}$ being a feasible solution of Problem S-CPOS is $\mathsf{Pr}\left\{\widetilde{F}(\hat{\boldsymbol{\theta}}_{\mathsf{bad}}, \mathcal{D}_{N}) \geq 1 - \tilde{\alpha}_{\mathsf{s}}^{\mathsf{trf}}\right\}$ which satisfies that

$$\begin{aligned} \mathsf{Pr}\left\{\widetilde{F}(\hat{\boldsymbol{\theta}}_{\mathsf{bad}}, \mathcal{D}_{N}) \geq 1 - \tilde{\alpha}_{\mathsf{s}}^{\mathsf{trf}}\right\} &= \mathsf{Pr}\left\{\widetilde{F}(\hat{\boldsymbol{\theta}}_{\mathsf{bad}}, \mathcal{D}_{N}) - F(\hat{\boldsymbol{\theta}}_{\mathsf{bad}}) \geq 1 - \alpha_{\mathsf{s}}^{\mathsf{trf}} + \alpha_{\mathsf{s}}^{\mathsf{trf}} - \tilde{\alpha}_{\mathsf{s}}^{\mathsf{trf}} - F(\hat{\boldsymbol{\theta}}_{\mathsf{bad}})\right\} \\ &\leq \mathsf{Pr}\left\{\widetilde{F}(\hat{\boldsymbol{\theta}}_{\mathsf{bad}}, \mathcal{D}_{N}) - F(\hat{\boldsymbol{\theta}}_{\mathsf{bad}}) \geq \alpha_{\mathsf{s}}^{\mathsf{trf}} - \tilde{\alpha}_{\mathsf{s}}^{\mathsf{trf}}\right\} \\ &= \mathsf{Pr}\left\{\left(\widetilde{F}(\hat{\boldsymbol{\theta}}_{\mathsf{bad}}, \mathcal{D}_{N}) - F(\hat{\boldsymbol{\theta}}_{\mathsf{bad}})\right)N \geq (\alpha_{\mathsf{s}}^{\mathsf{trf}} - \tilde{\alpha}_{\mathsf{s}}^{\mathsf{trf}})N\right\} \\ &= \mathsf{Pr}\left\{\left(\sum_{i=1}^{N}Y_{i} - \mathbb{E}\{Y_{i}\}\right) \geq (\alpha_{\mathsf{s}}^{\mathsf{trf}} - \tilde{\alpha}_{\mathsf{s}}^{\mathsf{trf}})N\right\}, \end{aligned} \tag{52}$$

where $Y_i$ is defined by
$$Y_i := 1 - \mathbb{I}(g(\mathbf{s}^{+,(i)})).$$
According to Hoeffding's inequality [23], (52) implies
$$\Pr\left\{\widetilde{F}(\hat{\boldsymbol{\theta}}_{\mathsf{bad}}, \mathcal{D}_N) \geq 1 - \tilde{\alpha}_{\mathsf{s}}^{\mathsf{trf}}\right\} \leq \exp\left\{-\frac{2N^2(\alpha_{\mathsf{s}}^{\mathsf{trf}} - \tilde{\alpha}_{\mathsf{s}}^{\mathsf{trf}})^2}{\sum_{i=1}^{N}(1-0)^2}\right\} = \exp\left\{-2N(\alpha_{\mathsf{s}} - \tilde{\alpha}_{\mathsf{s}})^2(1 - \gamma_{\mathsf{unsafe}})^2\right\}.$$
(53)

Here, (53) means that the probability of $\hat{\boldsymbol{\theta}}_{\mathsf{bad}}$ being a feasible solution of Problem S-CPOS is smaller than $\exp\left\{-2N(\alpha_{\mathsf{s}} - \tilde{\alpha}_{\mathsf{s}})^2(1 - \gamma_{\mathsf{unsafe}})^2\right\}$, which implies that a feasible solution of Problem S-CPOS has a probability larger than $1 - \exp\left\{-2N(\alpha_{\mathsf{s}} - \tilde{\alpha}_{\mathsf{s}})^2(1 - \gamma_{\mathsf{unsafe}})^2\right\}$ to be a feasible solution of Problem CPOS. The optimal solution $\tilde{\boldsymbol{\theta}}_{\alpha_{\mathsf{s}}}(\mathbf{s}, \boldsymbol{\theta}_k, \mathcal{D}_N)$ of Problem S-CPOS is also included.

When $\tilde{\boldsymbol{\theta}}_{\alpha_{\mathsf{s}}}(\mathbf{s}, \boldsymbol{\theta}_k, \mathcal{D}_N)$ is feasible for Problem CPOS, it is feasible for Problem CLPS. By applying Theorem 5, we have that $\tilde{\boldsymbol{\theta}}_{\alpha_{\mathsf{s}}}(\mathbf{s}, \boldsymbol{\theta}_k, \mathcal{D}_N)$ is also feasible for Problem LPS and thus the policy admitted by $\tilde{\boldsymbol{\theta}}_{\alpha_{\mathsf{s}}}(\mathbf{s}, \boldsymbol{\theta}_k, \mathcal{D}_N)$ satisfies the joint chance constraint in Problem CCRL. □

## J  Conservative approximation by affine chance constraint

Firstly, we will show that an affine chance constraint exists to approximate the joint chance constraint conservatively. The approximate problem also has a flipping-based policy in the optimal solution set. Since the problem with affine chance constraint can be transformed into the generalized safe exploration (GSE) problem [42], MASE and shielding methods [2, 28] can be applied to solve it, which gives the approximate solution of Problem CCRL.

Let $H_{\mathsf{acc}}(\boldsymbol{\theta}, \mathbf{s})$ be a unsafety function defined by
$$H_{\mathsf{acc}}(\boldsymbol{\theta}, \mathbf{s}) := \mathbb{E}_{\boldsymbol{\tau}_\infty \sim \Pr_{\mathbf{s}_0,\infty}^{\boldsymbol{\pi}_{\boldsymbol{\theta}}}}\left\{\sum_{i=1}^{T}\mathbb{I}(g(\mathbf{s}_i)) \mid \mathbf{s}_0 = \mathbf{s}\right\}.$$
(54)

Notice that $H_{\mathsf{acc}}(\boldsymbol{\theta}, \mathbf{s})$ satisfies
$$H_{\mathsf{acc}}(\boldsymbol{\theta}, \mathbf{s}) = 1 - \sum_{i=1}^{T}\Pr_{\mathbf{s},\infty}^{\boldsymbol{\pi}_{\boldsymbol{\theta}}}\left\{\mathbf{s}_i \in \mathbb{S} \mid \mathbf{s}_0 = \mathbf{s}\right\}.$$

Thus, the constraint $H_{\mathsf{acc}}(\boldsymbol{\theta}, \mathbf{s}) \leq \varphi$ is equivalent to
$$\sum_{i=1}^{T}\Pr_{\mathbf{s},\infty}^{\boldsymbol{\pi}_{\boldsymbol{\theta}}}\left\{\mathbf{s}_i \in \mathbb{S} \mid \mathbf{s}_0 = \mathbf{s}\right\} \geq 1 - \varphi,$$

which is a special case of affine chance constraint [13]. The theorem of conservative approximation of joint chance constraint based on affine chance constraint is as follows:

**Theorem 9.** *Suppose that $\mathbb{S}$ is compact and Assumption 2 holds. Define a function $C_{\mathsf{acc}}(\boldsymbol{\theta}, \mathbf{s})$ as follows:*
$$C_{\mathsf{acc}}(\boldsymbol{\theta}, \mathbf{s}, \varphi) := H_{\mathsf{acc}}(\boldsymbol{\theta}, \mathbf{s}) - \varphi.$$
(55)
*If $\varphi \leq \alpha$, $C_{\mathsf{acc}}(\boldsymbol{\theta}, \mathbf{s}, \varphi)$ is a conservative approximation of joint chance constraint (3).*

*Proof.* From the definition of $H_{\mathsf{acc}}(\boldsymbol{\theta}, \mathbf{s})$ as (54), we have
$$H_{\mathsf{acc}}(\boldsymbol{\theta}, \mathbf{s}) = \mathbb{E}_{\boldsymbol{\tau}_\infty \sim \Pr_{\mathbf{s}_0,\infty}^{\boldsymbol{\pi}_{\boldsymbol{\theta}}}}\left\{\sum_{i=1}^{T}\mathbb{I}(g(\mathbf{s}_i)) \mid \mathbf{s}_0 = \mathbf{s}\right\} = \sum_{i=1}^{T}\mathbb{E}_{\boldsymbol{\tau}_\infty \sim \Pr_{\mathbf{s}_0,\infty}^{\boldsymbol{\pi}_{\boldsymbol{\theta}}}}\left\{\mathbb{I}(g(\mathbf{s}_i)) \mid \mathbf{s}_0 = \mathbf{s}\right\}$$
$$= \sum_{i=1}^{T}\mathbb{E}_{\boldsymbol{\tau}_\infty \sim \Pr_{\mathbf{s}_0,\infty}^{\boldsymbol{\pi}_{\boldsymbol{\theta}}}}\left\{\mathbb{I}(g(\mathbf{s}_i)) \mid \mathbf{s}_0 = \mathbf{s}\right\} = \sum_{i=1}^{T}\Pr_{\mathbf{s},\infty}^{\boldsymbol{\pi}_{\boldsymbol{\theta}}}\left\{\mathbf{s}_i \notin \mathbb{S} \mid \mathbf{s}_0 = \mathbf{s}\right\}.$$
(56)

Due to Boole's inequality (p. 14 of [10]), we further have
$$H_{\mathsf{acc}}(\boldsymbol{\theta}, \mathbf{s}) = \sum_{i=1}^{T}\Pr_{\mathbf{s},\infty}^{\boldsymbol{\pi}_{\boldsymbol{\theta}}}\left\{\mathbf{s}_i \notin \mathbb{S} \mid \mathbf{s}_0 = \mathbf{s}\right\} \geq \Pr_{\mathbf{s},\infty}^{\boldsymbol{\pi}_{\boldsymbol{\theta}}}\left\{\mathbf{s}_i \notin \mathbb{S}, \forall i \in [T] \mid \mathbf{s}_0 = \mathbf{s} \in \mathbb{S}\right\}.$$
(57)

Thus, by (57), the following holds

$$H_{\text{acc}}(\boldsymbol{\theta}, \mathbf{s}) - \varphi \leq 0 \Rightarrow \Pr_{\mathbf{s},\infty}^{\pi_{\boldsymbol{\theta}}} \{\mathbf{s}_i \notin \mathbb{S}, \forall i \in [T] \mid \mathbf{s}_0 = \mathbf{s} \in \mathbb{S}\} \leq \varphi, \qquad (58)$$

Since the left side of (58) is equivalent to

$$H_{\text{acc}}(\boldsymbol{\theta}, \mathbf{s}) - \varphi \leq 0 \Rightarrow \Pr_{\mathbf{s},\infty}^{\pi_{\boldsymbol{\theta}}} \{\mathbf{s}_i \in \mathbb{S}, \forall i \in [T] \mid \mathbf{s}_0 = \mathbf{s} \in \mathbb{S}\} \geq 1 - \varphi,$$

(58) implies that $C_{\text{acc}}(\boldsymbol{\theta}, \mathbf{s}, \varphi)$ is a conservative approximation of joint chance constraint (3). $\qquad \square$

MDP with affine chance constraint can be written by

$$\max_{\boldsymbol{\pi} \in \Pi} \quad V^{\boldsymbol{\pi}}(\mathbf{s})$$
$$\text{s.t.} \quad \sum_{i=1}^{T} \Pr_{\mathbf{s}_0,\infty}^{\boldsymbol{\pi}} \{\mathbf{s}_{k+i} \notin \mathbb{S} \mid \mathbf{s}_k \in \mathbb{S}\} \leq \alpha, \ \forall k = 0, 1, 2, ..., \qquad (\mathsf{A}_\alpha(\mathbf{s}))$$

where the constraint of Problem $\mathsf{A}_\alpha(\mathbf{s})$ is a conservative approximation of (1), which can be proved following the same flow of Theorem 9. The following theorem for Problem $\mathsf{A}_\alpha(\mathbf{s})$ holds:

**Theorem 10.** *A flipping-based policy exists in the optimal solution set of Problem $\mathsf{A}_\alpha(\mathbf{s})$.*

*Proof.* Define a function $\mathbb{H}_{\text{acc}}^\star(\mathbf{s}, \mathbf{a})$ by

$$\mathbb{H}_{\text{acc}}^\star(\mathbf{s}, \mathbf{a}) = \sum_{i=1}^{T} \Pr_{\mathbf{s},\infty}^{\pi_{\text{acc}}^\star} \{\mathbf{s}_{k+i} \notin \mathbb{S} \mid \mathbf{s}_k = \mathbf{s}, \mathbf{a}_k = \mathbf{a}\}. \qquad (59)$$

Here, $\pi_{\text{acc}}^\star$ is an optimal solution of Problem $\mathsf{A}_\alpha(\mathbf{s})$. The continuity of $\mathbb{H}_{\text{acc}}^\star(\mathbf{s}, \mathbf{a})$ is guaranteed by Assumption 2 and the continuity of $g(\cdot)$ (pp. 78-79 of [25]). With $\mathbb{H}_{\text{acc}}^\star(\mathbf{s}, \mathbf{a})$, a probability measure optimization problem is defined as follows:

$$\max_{\boldsymbol{\mu} \in M(\mathcal{A})} \quad \int_{\mathcal{A}} Q_\alpha^\star(\mathbf{s}, \mathbf{a}) \, \mathrm{d}\boldsymbol{\mu}$$
$$\text{s.t.} \quad \int_{\mathcal{A}} \mathbb{H}_{\text{acc}}^\star(\mathbf{s}, \mathbf{a}) \, \mathrm{d}\boldsymbol{\mu} \leq \alpha. \qquad (\mathsf{B}_\alpha^{\text{acc}}(\mathbf{s}))$$

By just repeating the proof of Theorem 1, we can obtain that the optimal objective value of Problem $\mathsf{B}_\alpha^{\text{acc}}(\mathbf{s})$ equals the one of Problem $\mathsf{A}_\alpha(\mathbf{s})$ for any $\mathbf{s} \in \mathcal{S}$. A flipping-based version of Problem $\mathsf{B}_\alpha^{\text{acc}}(\mathbf{s})$ is written by

$$\max_{\mathbf{a}_{(1)}, \mathbf{a}_{(2)}, w} \quad w Q_\alpha^\star(\mathbf{s}, \mathbf{a}_{(1)}) + (1-w) Q_\alpha^\star(\mathbf{s}, \mathbf{a}_{(2)})$$
$$\text{s.t.} \quad w \mathbb{H}_{\text{acc}}^\star(\mathbf{s}, \mathbf{a}_{(1)}) + (1-w) \mathbb{H}_{\text{acc}}^\star(\mathbf{s}, \mathbf{a}_{(2)}) \geq 1 - \alpha. \qquad (\mathsf{W}_\alpha^{\text{acc}}(\mathbf{s}))$$

Since the continuity of $\mathbb{H}_{\text{acc}}^\star(\mathbf{s}, \mathbf{a})$ holds and it is bounded within $[0, 1]$, Theorem 10 can be proved by following the same process of proving Theorem 2 after replacing $\mathbb{P}^\star(\mathbf{s}, \mathbf{a})$ by $\mathbb{H}_{\text{acc}}^\star(\mathbf{s}, \mathbf{a})$. $\qquad \square$

## K  Details of Numerical Example

We present the details of our numerical example. The system dynamics is described by

$$\begin{bmatrix} x_{k+1} \\ y_{k+1} \end{bmatrix} = \begin{bmatrix} 1 & 0 \\ 0 & 1 \end{bmatrix} \begin{bmatrix} x_k \\ y_k \end{bmatrix} + \mathrm{d}t \begin{bmatrix} u_k + \delta_k \\ v_k + \zeta_k \end{bmatrix}.$$

Here, $\mathrm{d}t$ is the sampling time, the system state is $\mathbf{s}_k := [x_k \ y_k]^\top$ representing the position of the point, the action is $\mathbf{a}_k := [u_k \ v_k]^\top$ representing the velocity on each direction, and the disturbance vector is $\mathbf{d}_k := [\delta_k \ \zeta_k]^\top$ representing the system disturbance. Both $\delta_k$, $\zeta_k$ are random variables with zero means and standard deviations as 0.6. The initial point is $\mathbf{s}_0 = [0 \ 0]^\top$. The goal point is $\mathbf{s}_g = [15 \ 15]^\top$. The instantanuous loss function at step $k$ is $\ell(\mathbf{s}_k) := \|\mathbf{s}_k - \mathbf{s}_g\|_2^2$. For a given time-horizon $T$, we consider the joint chance constraint $\Pr \left\{ \left( \wedge_{k=1}^{T} s_k \notin \mathcal{O}_1 \right) \wedge \left( \wedge_{k=1}^{T} s_k \notin \mathcal{O}_2 \right) \right\} \geq 1 - \alpha$, where the dangerous regions $\mathcal{O}_1$ and $\mathcal{O}_2$ are defined by $\mathcal{O}_1 := \{\mathbf{s} : \|\mathbf{s} - \mathbf{s}_{\text{o1}}\|_2 \leq 2.5\}$, $\mathbf{s}_{\text{o1}} = [7.5 \ 10]^\top$

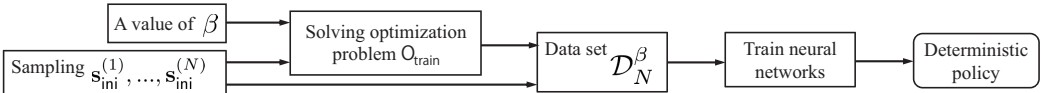

Figure 6: The framework of heuristically obtaining the optimal deterministic policy.

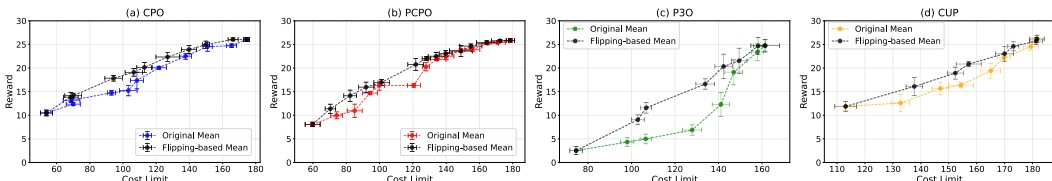

Figure 7: Experimental results on Safety Gym (PointGoal2). The flipping-based policy improves the performance of (a) CPO, (b) PCPO, (c) P3O, and (d) CUP. Error bars represent $1\sigma$ confidence intervals across 5 (CPO, PCOP) or 3 (P3O, CUP) different random seeds.

and $\mathcal{O}_2 := \left\{ \mathbf{s} : \|\mathbf{s} - \mathbf{s}_{\mathsf{o}2}\|_2 \leq 2.5 \right\}$, $\mathbf{s}_{\mathsf{o}2} = [10\ 5]^\top$. This numerical example investigates whether and when optimal flipping-based policy outperforms optimal deterministic policy under the same violation probability constraint. Therefore, instead of validating the algorithms of optimizing the policy, we implemented a heuristic method to obtain an optimal deterministic policy for each violation probability limit $\alpha$. Then, following Algorithm 1, we obtained the optimal flipping-based policy for each $\alpha$. The framework of heuristically obtaining the optimal deterministic policy is summarized in Figure 6. The heuristic method to obtain an optimal deterministic policy includes two steps. First, for a given initial state $\mathbf{s}_{\mathsf{ini}}^{(i)}$, we solve the following optimization problem ($\mathsf{O}_{\mathsf{train}}$):

$$
\begin{aligned}
\min_{\mathbf{a}_0, \dots, \mathbf{a}_{T-1}} \quad & \sum_{k=1}^{T} \ell(\mathbf{s}_k) \\
\text{s.t.} \quad & \mathbf{s}_{k+1} = \mathbf{A}\mathbf{s}_k + \mathsf{d}t(\mathbf{a}_k + \mathbf{d}_k),\ \mathbf{s}_0 = \mathbf{s}_{\mathsf{ini}}^{(i)} \\
& s_k \notin \widetilde{\mathcal{O}}_{1,k}^{\mathsf{ext}},\ s_k \notin \widetilde{\mathcal{O}}_{2,k}^{\mathsf{ext}},\ \forall k = 1, \dots, T.
\end{aligned}
\tag{$\mathsf{O}_{\mathsf{train}}$}
$$

Here, $\mathbf{A}$ is a two-dimensional identity matrix and the extended dangerous regions $\widetilde{\mathcal{O}}_1^{\mathsf{ext}}$ and $\widetilde{\mathcal{O}}_2^{\mathsf{ext}}$ are defined by $\widetilde{\mathcal{O}}_{1,k}^{\mathsf{ext}} := \left\{ \mathbf{s} : \|\mathbf{s} - \mathbf{s}_{\mathsf{o}1}\|_2 \leq 2.5 + 0.6\beta\sqrt{k}\mathsf{d}t \right\}$, $\mathbf{s}_{\mathsf{o}1} = [7.5\ 10]^\top$ and $\widetilde{\mathcal{O}}_{2,k}^{\mathsf{ext}} := \left\{ \mathbf{s} : \|\mathbf{s} - \mathbf{s}_{\mathsf{o}2}\|_2 \leq 2.5 + 0.6\beta\sqrt{k}\mathsf{d}t \right\}$, $\mathbf{s}_{\mathsf{o}2} = [10\ 5]^\top$. Here, $\beta$ is a coefficient to regulate the violation probability. Note that the disturbance obeys Gaussian distribution with zero covariance and the same deviation, and the confidence region for any given probability can be described by a circle. Thus, by regulating $\beta$, we can ensure the probability confidence of the obtained solution. For a given $\beta$, if we solve problem ($\mathsf{O}_{\mathsf{train}}$) for any $\mathbf{s}_{\mathsf{ini}}^{(i)}, i = 1, \dots, N$ and extract the first one $\hat{\mathbf{a}}_0^{(i)}$ of the solution sequence, we can obtain a set $\mathcal{D}_N^\beta := \left\{ (\mathbf{s}_{\mathsf{ini}}^{(i)}, \hat{\mathbf{a}}_0^{(i)}) \right\}_{i=1}^{N}$. Then, we can use $\mathcal{D}_N^\beta$ to train a neural network-based policy with the state as input and the action as output. We obtain neural network-based policies with different violation probability thresholds by varying $\beta$ from 1 to 2.2 with 0.05 as an increment. In the test, we use the inverse distance as a metric for evaluating performance, which is defined by $r(\mathbf{s}_k) := 1/\left( \|\mathbf{s}_k - \mathbf{s}_{\mathsf{g}}\|_2^2 + 0.1 \right)$. We tested each neural network with five times simulation sets. In each simulation set, one thousand simulations were conducted to calculate the violation probability and the mean reward.

## L   Details of Safety-Gym Experiment

We used a machine with Intel(R) Core(TM) i7-14700 CPU, 32GB RAM, and NVIDIA 4060 GPU. We present the details of our experiments using Safety Gym. Our experimental setup differs slightly from the original Safety Gym in that we deterministically replace the obstacles (i.e., unsafe regions). This modification ensures that the environment is solvable and that a viable solution exists. Details of our experiments using Safety Gym are as follows. To ensure the generalization of the algorithm, we

Table 1: Hyper-parameters for Safety Gym experiments.

| | NAME | VALUE |
|---|---|---|
| | NETWORK ARCHITECTURE | $[64, 64]$ |
| | ACTIVATION FUNCTION | tanh |
| | LEARNING RATE (CRITIC) | $2 \times 10^{-4}$ |
| | LEARNING RATE (POLICY) | $3 \times 10^{-3}$ |
| | LEARNING RATE (PENALTY) | 0.0 |
| COMMON PARAMETERS | DISCOUNT FACTOR (REWARD) | 0.99 |
| | DISCOUNT FACTOR (SAFETY) | 0.995 |
| | STEPS PER EPOCH | $40,000$ |
| | NUMBER OF CONJUGATE GRADIENT ITERATIONS | 20 |
| | NUMBER OF ITERATIONS TO UPDATE THE POLICY | 10 |
| | NUMBER OF EPOCHS | 500 |
| | TARGET KL | 0.01 |
| | BATCH SIZE FOR EACH ITERATION | 1024 |
| CPO & PCPO | DAMPING COEFFICIENT | 0.1 |
| | CRITIC NORM COEFFICIENT | 0.001 |
| | STD UPPER BOUND, AND LOWER BOUND | $[0.425, 0.125]$ |
| | LINEAR LEARNING RATE DECAY | TRUE |

Table 2: Test settings for Safety Gym experiments.

| | NAME | VALUE |
|---|---|---|
| | DAMPING COEFFICIENT | 0.1 |
| TEST SETTING | STEPS PER EPOCH | $10,000$ |
| | NUMBER OF EPOCHS | 60 |

Note: Same training parameters as in training process except for training scale.

used the original SafetyPointGoal2-v0 environment. In the initial stage, we identified parameters with good convergence properties according to specified criteria. Using these parameters, we trained the CPO and PCPO algorithms at 10 cost limit intervals within the range of 40-180, continuing until the policy network converged. In the testing experiments, we utilized the saved parameters from these converged networks, loading them into the policy network and sampling data under different seeds. Finally, we selected the policy network at an appropriate cost limit as the base network for the Flipping-based policy. When the two base networks output different actions at each step, we chose different actions according to the flip probability, sampling the results under various seed environments to obtain the final results for the Flipping-based policy.

**Collision Probability Analysis.** We monitored whether the agent encountered collisions under each single-step condition across all tests. Subsequently, we computed the collision probabilities for $T = 3, 10, 30$ by assessing the presence of collisions across every $T$ consecutive step. The findings are presented in Figure 4. It is important to acknowledge that, despite utilizing a trained and converged stable policy network during the collision testing phase, the collision probability and

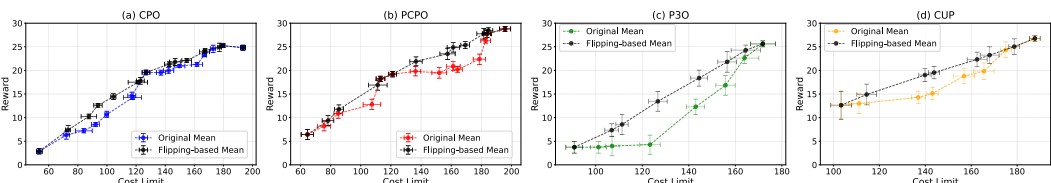

Figure 8: Experimental results on Safety Gym (CarGoal2). The flipping-based policy improves the performance of (a) CPO, (b) PCPO, (c) P3O, and (d) CUP. Error bars represent $1\sigma$ confidence intervals across 3 different random seeds.

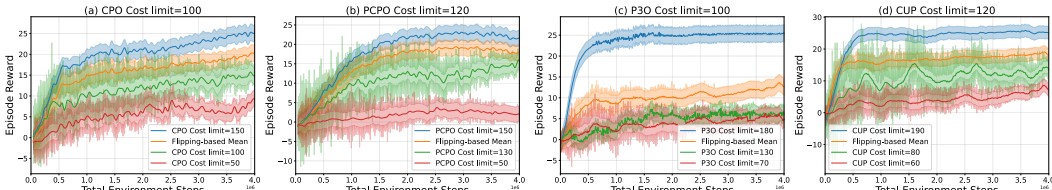

Figure 9: Experimental results on Safety Gym (PointGoal2). Reward profiles during the training processes of (a) CPO, (b) PCPO, (c) P3O, and (d) CUP. Error bars represent $1\sigma$ confidence intervals across 5 (CPO, PCOP) or 3 (P3O, CUP) different random seeds.

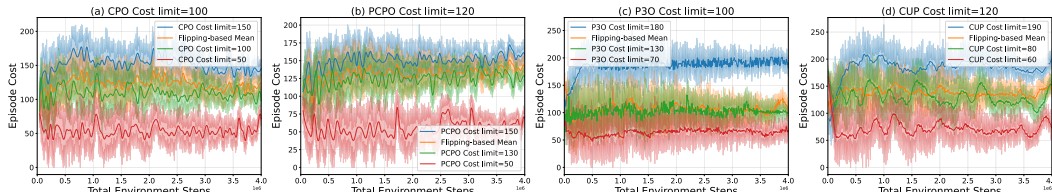

Figure 10: Experimental results on Safety Gym (PointGoal2). Cost profiles during the training processes of (a) CPO, (b) PCPO, (c) P3O, and (d) CUP. Error bars represent $1\sigma$ confidence intervals across 5 (CPO, PCOP) or 3 (P3O, CUP) different random seeds.

cost limit curves might exhibit some instability, attributed to the relatively small sample size of 25 trajectories. This variability is likely because the CPO and PCPO algorithms were not primarily designed to minimize collision probabilities. Nonetheless, the curves demonstrate a clear positive correlation, affirming the validity and reliability of our experimental outcomes. Our analysis further supports that the flipping-based method effectively enhances reward performance without increasing collision risks.

We also conducted experiments for two more baselines, P3O and CUP on PointGoal2. Figure 7 presents the experimental results, showing that the flipping-based policy can also enhance the performance of P3O and CUP. Additionally, experiments were conducted for the four baselines—CPO, PCPO, P3O, and CUP—under the CarGoal2 environment, with the results summarized in Figure 8. The findings in CarGoal2 are consistent with those in PointGoal2.

Figures 9 and 10 summarize reward and cost profiles of the training processes for each baseline algorithm on PointsGoal2. The training results further confirm that combining a performance policy with a safe policy yields a flipping-based policy that outperforms the policy trained by the original algorithm, while adhering to the required cost limit.

