# OpenReview forum: "Flipping-based Policy for Chance-Constrained Markov Decision Processes"
_NeurIPS.cc/2024/Conference — NeurIPS 2024 poster_

### Official Review · Reviewer_7QvG · 2024-07-05

**Soundness:** 3
**Presentation:** 2
**Contribution:** 2
**Rating:** 7
**Confidence:** 3

**Summary:**

The paper is dealing with a chance constrained MDP problem, where the authors constrain the probability of staying in the safe set throughout an episode. The authors show that the original problem can be recast using a flipping-based policy, where the optimal decision to satisfy the constraint is conditioned on the current state (regardless if it is in the safe set or not). The authors proceed to develop a practical algorithm that can be applied to standard RL algorithms such as CPO and demonstrate the utility of their approach. In particular they show that the flipping based policy can outperform significantly a deterministic policy

**Strengths:**

1. The paper is rigorous and the maths seem to be correct, however, I have a few questions and clarifications and to be honest I didn’t have time to validate the proofs.
1. The “secret” of many practical safe RL research is that, in fact, that many algorithms are shown to train well a deterministic policy. The authors demonstrate that there is a possibility to train a truly stochastic policy as well. I find this line of research very valuable.

**Weaknesses:**

1. Some of the theoretical results would benefit from explanations and intuitions behind them. For instance, why do they work? See questions
1. I found the discussion of the practical algorithm confusing (but keep in mind time limitation for reviews). For example, the problem LP is not clear. See questions.
1. The paper is heavy on the theory side, which is not necessarily a weakness, but the experimental side could be improved. For example, I am not sure how, in fact, easy it is to adapt, e.g., safe PPO to a flipping-based policy.

**Questions:**

1. Proposition 1 seems like a very strong result and I don’t have access [21]. Could the authors restate Theorem 1.3 from [21] and explain the intuition behind it. For example, how is it possible to cast an infinite dimensional problem to finite dimensions?
1. What is the intuition behind the possibility of using only two policies and not L as Proposition 1 suggests?
1. If I understand correctly the problem LP solves a number of PDPRL problems for a number of different safety levels $\alpha_i$ and then compares it with a predefined safety level $\alpha$. Then chooses only two values of $\alpha_i$ for the final flipping policy. Could you explain the intuition behind it. Why does this approach work?
1. What steps one has to take to adapt say safe PPO to a flipping-based policy?
1. How did you reformulate the PointGoal problem as a chance constrained problem?

**Limitations:**

limitations are discussed

---

> ### Author Rebuttal · Authors · 2024-08-05
>
> We appreciate the reviewer's encouraging comments. Your feedback and suggestions are valuable and help us improve the quality of our manuscript. All responses are given in a point-to-point way.
>
> **Proposition 1 [Question 1 and Weakness 1].**
> We will restate Theorem 1.3 from [21]. We proved a weak version by ourselves and will present it to help the reviewer understand it.
>
> First, we restate Theorem 1.3 of [21], which gives the following two results about Problem PMO:
> (a) The optimal solution to Problem PMO exists;
> (b) An optimal solution is a finite linear combination of Dirac measures.
>
> The result (b) implies a discrete probability measure in the optimal solution set of Problem PMO. It does not give the exact number $L$ of the finite dimension, which can be extremely close to infinite but countable. It does not clarify which points should be given the discrete probability measure. The points and the weights of linear combinations should be optimized.
>
> Then, we explain the intuition behind Theorem 1.3 of [21] by introducing a sketch of how we prove a weak version of it. The existence of the optimal solution can be easily proved using the Prokhorov Theorem. Here, we focus on the weak version of result (b). We prove that Problem $B_\alpha(s, L)$'s optimal value converges to Problem PMO's optimal value when $L\rightarrow\infty.$ Since Problem $B_\alpha(s, L)$'s feasible set is a subset of Problem PMO for any $L$, Problem $B_\alpha(s, L)$'s optimal value should be no more than the one of Problem PMO. If Problem $B_\alpha(s, L)$'s optimal value converges to a value equal to or larger than Problem PMO's optimal value, the proof is done. We can show that there will be a sequence of measures converging to the optimal solution of Problem PMO from inside by using Prokhorov Theorem. All the measures in the sequence are feasible and not on feasible region's boundary. Each measure in the sequence can be approximated by finite measures. Each measure in the sequence has the objective value as the limit of finite measures's objective value. These finite measures are feasible for Problem $B_\alpha(s, L)$ and should be smaller than Problem $B_\alpha(s, L)$'s optimal value. Thus, in the sequence, each measure's objective value is smaller than Problem $B_\alpha(s, L)$'s optimal value. Therefore, the limit, Problem PMO's optimal value, should be smaller than Problem $B_\alpha(s, L)$'s optimal value, which completes the proof.
>
> **Why only two policies [Questions 2 and 3, Weakness 2].**
> Based on Theorem 1, Propositions 1 and 2, we proved Theorem 2. **Proposition 2 is the first transition from Proposition 1 to Theorem 2.** Proposition 1 suggests that $L$ policies should be enough. Proposition 2 claims that another problem (Problem $V_\alpha(s, L)$) that optimizes the discrete probability measure of violation probability's interval has the same optimal objective value as Problem $B_\alpha(s, L)$ for any $L$. We further can show that Problem $V_\alpha(s, L)$ has the optimal value whose negative is Problem $H_\alpha(s)$'s optimal value for any given $L$. **Problem $H_\alpha(s)$ is a linear program in a two-dimension plane, which is the second transition from Proposition 1 to Theorem 2.** Intuitively, we use some geometric properties (Supporting hyperplane theorem and Caratheodory’s theorem) of the solution Problem $H_\alpha(s)$ to prove that Problem $V_\alpha(s, L)$'s optimal value does not increase when $L\geq 2.$ Which is equivalent to say that Problem $B_\alpha(s, L)$'s optimal value does not increase when $L\geq 2.$ Note that Proposition 1 says that $L$ is enough to get optimality for Problem PMO and we show $L\geq 2$ does not increase the objective value. Thus, two policies are enough. The same process can be repeated to explain Question 3. Problem PSPRL is a special case of Problem PMO, and Problem PFPRL is a special case of Problem $V_\alpha(s, L)$. Thus, we have Theorem 6 to show that Problem PFPRL and Problem PSPRL share the same optimal value. Then, Problem PSPRL can attain the optimality by two policies. Note that Problem LP's feasible region is a subset of the one of Problem PSPRL. Problem PSPRL considers all possible measures defined on the interval $[0,1]$. Problem LP only considers discrete measures defined on a finite subset of $[0,1]$. Since Problem PSPRL attains the optimality with two policies by Theorem 6, Problem LP also attains the optimality with two policies as its special case. **Since Problem LP only considers finite samples from the interval $[0,1]$ and these finite samples are randomly chosen instead of being optimized, thus we can only show that the optimal value of Problem LP converges to the one of Problem PSPRL with probability 1 when $L\rightarrow\infty$**.
>
> **Adapt Safe PPO [Question 4 and Weakness 3].**
> Thanks for your question. Adapting safe PPO to a flipping-based policy includes two steps. The first step is to use Algorithm 1 (Page 7 of our manuscript) to train the flipping-based policy using safe PPO in step 2.
> Then, implement the flipping-based policy according to Algorithm 2 (Page 7 of our manuscript). We added P3O [Ref-4-1] into the comparison, which we think is an effective safe PPO method. We also conducted additional experiments in another environment (Cargo) to validate all the methods. In any case, using a flipping-based policy can improve the expected reward under the same expected cost. The new results are summarized in a new One-Page PDF file attached to the global response. Please kindly check the One-Page PDF file.
> We will add the new results in the new version.
>
> **PointGoal Chance Constrained Problem [Question 5].**
> The chance constraint for the PointGoal is that the collision probability in the future $T$ steps should be less than $\alpha$.
>
> [Ref-4-1] Zhang, L., et al, Penalized proximal policy optimization for safe reinforcement learning. In IJCAI, 2022.

---

> > ### Comment · Reviewer_7QvG · 2024-08-11
> >
> > Thank you for a detailed response.
> >
> > Re $L = 2$. If I followed your explanation, the fact that we use chance constraint with one parameter $\alpha$ implies that we need to policies to find the optimal value of $V$. Could that imply that in others more complex problem formulations, we may need to choose more than one policy?

---

> > > ### Author Response · Authors · 2024-08-12
> > > **Thank you for your additional comments**
> > >
> > > We appreciate your valuable advice and comments. Thank you for increasing the score! We ensure that all reviewers’ feedback will be reflected in the new version of our paper. The question about the number of policies in the official comment is very important and interesting. According to our recent results, more complex problem formulations will need more policies. We would like to share the conclusion about the expected cumulative safety constraints. The number of necessary policies equals the number of constraints plus one. Namely, we have $l=m+1$, where $l$ is the number of necessary policies and $m$ is the number of expected cumulative safety constraints. It also works for the chance constraint in this manuscript since the joint chance constraint is essentially one constraint for the probability of an event.
> > >
> > > Thank you again for your patience and time for our manuscript!

---

> > > > ### Comment · Reviewer_7QvG · 2024-08-12
> > > >
> > > > Many thanks for the response! I apologise for the typos, but you did answer my question!
> > > >
> > > > I think it's an interesting observation that the degrees of freedom in the number of policies ($L$) is related to the problem complexity. It seems that in the considered case one needs two policies dealing with the constraint violations (almost like a performance policy and a safety policy). If an additional insight can be drawn from your experiments, I think it's worth adding a few lines on to the final version of the paper.

---

> > > > > ### Author Response · Authors · 2024-08-12
> > > > > **Reply to the Official Comment by Reviewer 7QvG**
> > > > >
> > > > > Thanks for your additional comment! It is helpful. We will add some descriptions of the additional insight in the new version. Thank you again for your time and comments!

---

### Official Review · Reviewer_7qnc · 2024-07-11

**Soundness:** 4
**Presentation:** 4
**Contribution:** 3
**Rating:** 7
**Confidence:** 3

**Summary:**

The paper presents a number of optimality results for flipping-based policies in safe reinforcement learning. It first establishes that flipping-based policies are overall optimal for joint chance constrained MDPs, which represents a significant reduction of the original optimization formulation. Then, acknowledging that even this formulation is practically intractable, it provides a number of relaxations and principled approximations to get to a computationally tractable procedure, and show that one can find flipping-based policies that outperform the best deterministic policies that one can find.

**Strengths:**

1. Safe RL is a very important and challenging problem.
2. Theorem 2 (and 4 and 6 and 7) is quite interesting and, at least to me, surprising!
3. The authors take a methodical and principled approach to connect their main theory in section 3 with a practical algorithm that can be used for RL by the end of section 4.

**Weaknesses:**

1. Theorems 5 and 8 don’t seem very strong or useful since they don’t specify the constraints on \gamma_unsafe and T (just say they exist), so in particular it may not hold at all for a given user-specified T, and even if it does, it doesn’t provide a way for the user to find a \gamma_unsafe that makes the approximation conservative for their specified T.

**Questions:**

1. line 91: are there \alpha subscripts missing in a couple places here? I don’t think, e.g., \pi^* has been defined without an \alpha subscript.
2. Line 204: I think there’s a conditioning on s_k=s missing in the definition of F^d. Or otherwise I don’t know what the s in the subscript means.
3. In 4.3, is there a distinction between \Theta and \boldsymbol{\Theta}?
4. In 5.2, I think you mean to say the reward INCREASED with the flipping policy
5. Theorem 6 seems far stronger than Theorem 2, and I feel this gap is quite counterintuitive and unacknowledged by the authors, though maybe I’m missing something; if this point could be clarified, I would raise my score. In particular, Theorem 2 says that in the current state, the controller can select from just two (state-dependent) actions (with a state-dependent flipping probability) at the current time point while remaining optimal. But Theorem 6 seems to say that the entire optimal policy can be written as a (fixed, not-state-dependent) mixture of two policies. Perhaps most surprising is the fact that the flipping weight between the two policies is not state-dependent, whereas in Theorem 2 it was allowed to depend on state. Is there an easy way to see why it’s not so surprising? For instance, if the parameterized deterministic policy class is a universal approximating (deterministic) function class, then the PSPRL optimum should essentially be the same as the original CCRL optimum, right (I can see that if PSPRL can implement any policy via \theta and then arbitrarily mix between them, and by having a continuous mixture of policies that can take the same action at some states and different actions at others, the PSPRL optimum can, e.g., implement a switching policy with a different weight for every state)? But then Theorem 6, if I’m reading it right, says there’s an optimal policy for the original problem with a switching weight that doesn’t depend on state at all! What am I missing?

**Limitations:**

The experimental results certainly support the rest of the paper, but are somewhat unsurprising, since they basically just show that there can be a mixture of two deterministic policies that beats the best deterministic policy—no surprise there since mixtures of two deterministic policies is a larger class than deterministic policies. And the actual flipping policy found in figure 2 seems a bit troubling to me—in practice would one want a policy that improves upon another safe policy by with some probability taking a marginally unsafe shortcut? For, e.g., a self-driving car, I would probably say no, but maybe this is a weakness of the joint chance constraint formulation, which only conditions on safety, but not the state itself, when demanding joint safety for the next T time steps. I have to agree that, for the problem as formulated, the flipping policy in Figure 2 is indeed better than the deterministic policy. Perhaps a bit more discussion of the problem formulation is merited, as opposed to just referencing other works that use the same formulation.

---

> ### Author Rebuttal · Authors · 2024-08-05
>
> We sincerely appreciate the reviewer's constructive comments and insightful questions. We feel the reviewer has been patient and given many suggestions to help us improve the quality of our manuscript and questions to guide us in reflecting on our research. Except for your professional advice, your patience and kind heart also mean a lot to us.
>
> We will answer the questions first and then address the weaknesses and limitations. All responses are given point-to-point.
>
> **Typos [Question 1, 2, 3, 4].**
> Thank you for pointing out the typos. We apologize for the typos, which made it difficult for you to read this manuscript. For the typo at line 91, indeed, $\alpha$ is missing in a couple of places. For the typo at line 204, it should include the conditional part and then take a probability integration on the initial state. We will explain the reason in the response **Theorem 6 [Question 5].** In 4.3, we intend to only use $\boldsymbol{\Theta}$ (with boldsymbol). In 5.2, exactly, we mean INCREASED.
> We will correct the typos in the new version.
>
> **Theorem 6 [Question 5].**
> We deeply appreciate this valuable comment. The content of this comment is about the core part of our theoretical contribution. We acknowledge this gap between Theorem 6 and Theorem 2 very much. First, let us explain why Theorem 6 and Theorem 2 have different results on the flipping probability. The difference is caused by **a subtle difference between Problem CCRL and Problem PSPRL**. Problem CCRL relies on a revised Bellman equation (Theorem 1) to deliver the optimal policy for every possible state in the state space. The joint chance constraint also needs to be satisfied for every possible state in the pointwise way. In Problem PSPRL, the expectations of reward and constraint also consider the initial state's probability distribution, which is a usual way in the safe RL community. They are not for a specific state. Thus, the obtained flip probability is not state-dependent. Namely, **the subtle difference is the joint chance constraint should be satisfied pointwisely in Problem CCRL while it only needs to be satisfied in a sense of mean value regarding the initial state in Problem PSPRL.** Of course, as the reviewer mentioned, when the PSPRL optimum is the same as the original CCRL optimum, we can conclude that Problem CCRL also has an optimal policy with state-independent switching weight. Then, we have to prove that, for safe RL with joint chance constraint or expected cumulative safety constraints, the policy delivered by the revised Bellman equation (pointwise) and the policy delivered by taking the expectations with the initial state's probability distribution is equivalent. For RL without constraints, they are equivalent. Maybe there is a way to find some conditions, for example, constraining the initial state in a compact set, and then find some equivalent reformations between pointwise constraint and expected constraint on the initial state distribution.
>
> **Theorem 5 and 8 [Weakness 1].**
> Thanks for pointing out this weakness. As the reviewer mentioned, Theorems 5 and 8 only show the existence but do not show an explicit way to choose $\gamma_{\mathsf{unsafe}}$ for specified $T.$ One important information is that, for specified $T,$ if $\gamma_{\mathsf{unsafe}}$ is increased, the approximation for safety can be more conservative. If a conservative safety is desired, we recommend to use a $\gamma_{\mathsf{unsafe}}$ that is close to $1$.
>
> **Experimental results [Limitations 1].**
> Thanks for the insightful and helpful comments.
> First, let us emphasize that our theoretical results show that **the mixture of two deterministic policies can achieve optimality**. We do not need a mixture of three or more deterministic policies. Even if a mixture of three or more deterministic policies is a larger class than a mixture of two deterministic policies, it cannot further improve the expected reward for Problem PMO.
>
> As the reviewer mentioned, it is necessary to discuss the problem formulation more, especially in the cases where joint chance constraint formulation makes sense. If we understand the reviewer's point correctly, Figure 2 seems troubling because a high threshold of violation probability will be unacceptable for fatal collisions and does not make sense in a self-driving car. That is why we chose the word "dangerous regions" instead of "obstacles." It is dangerous but not fatal. Namely, we agree with the point that using joint chance constraints with a relatively high threshold is more suitable for some "soft" constraints but not fatal, for example, the risk of meeting traffic jams in the path planning of express delivery services, battery capacity in energy systems with renewable energy. For fatal constraints, safety should almost surely be guaranteed. Then, according to Theorem 3, a deterministic policy should be enough if it can be exactly found. However, for enhancing exploration, computing policy gradients, learning to make decisions, and other reasons, a stochastic policy is still needed for practical applications.
>
> Besides, Theorem 4 shows that the flipping-based policy can achieve optimality for expected cumulative safety constraints, which is generally used in safe RL. Although Theorem 4 uses the indication function and probability level to show the connection between Problem ECRL and Problem CCRL, the theoretical result can be directly adapted to the general expected cumulative safety constraints, which can extend our theory to more practical scenarios. We should emphasize this by a remark after Theorem 4 in the new version.
>
> Thank you again for your constructive comments.

---

> > ### Comment · Reviewer_7qnc · 2024-08-10
> >
> > Thm 6 / Q5: This is very helpful, thank you for the clarification! I would say that, even with your response and going back through the paper, it took me a while to see this subtle (but apparently important) distinction between CCRL and PSPRL--maybe the authors could try to highlight it more?
> >
> > For your experimental results--I agree with the bolded statement about your *theoretical* results, and indeed this is a surprising and nice result precisely because one's baseline expectation is that a mixture of more deterministic policies should be better than a mixture of fewer deterministic policies! My point was that your *empirical* results only confirm this expectation for one versus a mixture of two deterministic polices (and hence are not very surprising), not the surprising part that mixing in *more* deterministic policies *doesn't* help.
> >
> > Your examples with "dangerous regions" are quite compelling--thank you for clarifying this!
> >
> > I have raised my score to a 7.

---

> > > ### Author Response · Authors · 2024-08-11
> > > **Thank you for your additional reply**
> > >
> > > We would like to express our sincere gratitude to the reviewer for reading through our responses and having raised the score. We believe that the valuable comments from Reviewer 7qnc are very helpful for us to improve our paper. Thank you very much for your considerable comments.
> > >
> > > We will add a short but clear remark in the new version to highlight the subtle distinction between Problem CCRL and Problem PSPRL.
> > >
> > > Solving Problem LP naturally gives us a solution that is a mixture of two deterministic policies.
> > > To show that a mixture of more deterministic policies doesn't help exactly, we think it is necessary to propose an algorithm to solve PSPRL directly instead of obtaining an approximate solution by solving Problem LP.
> > > We leave this as future work. It is a very interesting point.
> > >
> > > Thank you again for your patience and helpful comments!

---

### Official Review · Reviewer_GDWH · 2024-07-13

**Soundness:** 3
**Presentation:** 2
**Contribution:** 2
**Rating:** 5
**Confidence:** 3

**Summary:**

The study proposes a flipping-based policy for managing chance constraints in Markov Decision Processes (MDPs). It introduces a probabilistic approach where actions are selected by flipping a "distorted coin", which is helpful in handling uncertainties in safety-critical environments. The authors establish a theoretical framework, including a Bellman equation adaptation for this setup and proofs of optimality under given constraints.

**Strengths:**

1. The flipping-based approach is a potentially impactful method for addressing uncertainty in decision-making processes.
2. The paper provides rigorous theoretical backing for the proposed method, including detailed proofs.

**Weaknesses:**

1. The algorithm involves multiple layers of optimization and probability adjustments which might be difficult to tune in real-world applications.
2. The experimental section primarily focuses on simulated environments (Safety Gym benchmarks), and the performance is not significantly improvement compared with CPO and PCPO. There are some new methods that significantly better than CPO and PCPO, e.g., https://arxiv.org/pdf/2405.05890 and https://arxiv.org/pdf/2405.01677

**Questions:**

1. Could we possibly use neural networks to predict action candidates and flip probabilities?
2. What is the difference between flip-based policy and reject sampling?

**Limitations:**

Yes

---

> ### Author Rebuttal · Authors · 2024-08-05
>
> We appreciate the reviewer's helpful comments and insightful questions. The questions and comments raised by the reviewer are addressed point-to-point as follows.
>
> **Neural networks for action and flip probability [Question 1].**
> Thank you for this good question! As the reviewer mentioned, it is possible to approximate the optimal flipping-based policy by using a neural network that takes the state as input and outputs the action candidates and flip probability. We can add Gaussian noises to each action candidate to introduce stochasticity and further enhance the exploration. We can also include the added Gaussian noises' covariances into the neural network output. Then, the neural network output becomes the parameters for a Gaussian mixture distribution. It becomes a mixture density network. We intend to train the flipping-based policy with the existing tools in safe reinforcement learning research's infrastructural frameworks, such as OmniSafe. In OmniSafe, the single Gaussian distribution-based stochastic policies can be trained directly without any adaptions. Thus, we have proposed practical algorithms (Section 4.3) to train the flipping-based policy directly using existing algorithms without adjustment. It is worth proposing a new efficient algorithm to train the mixture density network for safe RL. Particularly, with the theory in this paper, the Gaussian mixture distribution only needs two Gaussian kernels to ensure optimality.
>
> **Flipping-based policy and Reject sampling [Question 2].**
> In flipping-based policy, similar to the general stochastic policy, reject sampling are usually used to generate the input. When using the exact flipping-based policy, the neural network outputs two action candidates and flip probability. Then, the implemented action is chosen from two action candidates by checking a random value generated by the uniform distribution within $[0,1]$. The first candidate is chosen if the generated value surpluses the flip probability value. Otherwise, the second candidate will be chosen. A more complicated case is using a mixture density network to approximate the flipping-based policy. Then, the implemented action will be decided in two ways. The first is to use reject sampling directly for the Gaussian mixture distribution. Or we can sample two samples from two Gaussian distributions and then repeat the process for the case of the exact flipping-based policy. However, sampling from the Gaussian distribution is often achieved by reject sampling.
>
> **Complexity of algorithm [Weakness 1].**
> Thanks for pointing out this weakness in our research. The current algorithm's weakness is that it uses multiple layers of optimization and probability adjustments. This limitation of our method (mentioned in the Appendix) may stimulate new work on designing efficient algorithms to obtain the flipping-based policy, in which the neural network outputs two action candidates and the flip probability. The flipping-based policy has a structure that is much simplified compared to the general stochastic policy, but it can still ensure optimality.
>
> **Compare with new methods [Weakness 2].**
> Thank you for the useful comments and the wonderful papers. Although the methods presented in the recommended papers are very interesting and useful, we do not find the toolbox in Omnisafe, and the time is limited for us to implement them. Instead, we added CUP [Ref-2-1] and P3O [Ref-2-2] into the comparison. We will first claim our main contributions and then explain why our choices support our main contributions. Although we do not include the recommended methods, we will add the references to the new version of our paper since they are excellent related work of safe RL.
>
> First, please let us emphasize that our main contributions include **proving the optimality of flipping-based policy and one practical algorithm to realize it**. The practical algorithm can be CPO, PCPO, or any other safe RL algorithm. How much flipping-based policy can improve the existing algorithm depends also on the original algorithm's performance. To show the generality of our proposed policy and practical algorithm, we add its applications to CUP and P3O. We conducted additional experiment results with CUP and P3O as well. We also added another environment (Cargo) to validate all methods. In any case, using a flipping-based policy can improve the expected reward at the same cost. We would like to ask the reviewer to see the new result in a new One-Page PDF file attached to the global response. We consider it more important to **show the performance of improving existing algorithms by flipping-based policy**. We will add the new figures in the new version.
>
> We sincerely thank the reviewer for taking the time to review our paper.
>
> [Ref-2-1] Yang, L., et al, Constrained update projection approach to safe policy optimization. In NeurIPS, 2022.
>
> [Ref-2-2]  Zhang, L., et al, Penalized proximal policy optimization for safe reinforcement learning. In IJCAI, 2022.

---

### Official Review · Reviewer_g2rG · 2024-07-23

**Soundness:** 2
**Presentation:** 3
**Contribution:** 2
**Rating:** 5
**Confidence:** 3

**Summary:**

This paper introduces a new policy called the flipping-based policy for Chance-Constrained Markov Decision Processes (CCMDPs), which is useful in safe reinforcement learning. The policy uses a coin flip to choose between two actions, depending on the state. The authors establish a Bellman equation for CCMDPs and show that this flipping-based policy can be part of the optimal solution. To provide a practical algorithm, they also demonstrate how joint chance constraints can be approximated into Expected Cumulative Safety Constraints (ECSCs). The paper presents a framework for integrating this policy into existing safe RL algorithms like CPO and PCPO, showing improvements in performance on Safety Gym benchmarks while maintaining safety constraints.

**Strengths:**

1. This paper presents an extensive detailed theoretical analysis, with the entire analytical and proof process vividly illustrated in Figure 1 and Figure 5.
2. The concept of employing a flipping policy is quite novel, offering a simple yet potent perspective to solve the problem PMO.
3. This paper delivers a comprehensive framework from theoretical analysis to practical general safety RL algorithms, providing mathematically insightful and practically effective solutions. The experimental section and visualizations are designed to complement the core theorems, ensuring a cohesive understanding.

**Weaknesses:**

1. The typo in the abstract: expected cumulative safety constraints (ESCSs) should be ECSCs.
2. The typo in Algorithm 2, line 2: Remove one of the redundant as in the statement.
3. The assumption regarding the reward and state transition function, as mentioned by the author, is not inherently natural or generalizable across various applications. This is particularly evident in decision-making environments within robotics and autonomous driving, where reward signals are often non-continuous, and state transitions can be abrupt. This limitation significantly constrains the potential impact of the paper.
4. The paper, as indicated by its title and abstract, aims to approximate the solution to CCRL problems with ECRL. However, the text frequently shifts focus to extending CCRL to ECRL for generalization, which only serves to complicate the narrative and blur the paper's primary objective. It is recommended that the author refocus the discussion on the application of ECRL to CCRL for a clearer and more coherent presentation.
5. The experimental section exclusively presents the results of the test process, which, although demonstrating positive outcomes, fails to provide any insights into the training process itself. It is crucial for the author to include at least some intermediate results from the training process to validate that the process does not incur significant overheads.
6. The utilization of a flipping-based policy introduces broader confidence intervals in the results, a concern that may be particularly relevant in safety RL applications.
7. The authors should ensure that at least two environments are considered in experiments for broader applicability.
8. This paper should add a related work section to compare and contextualize the current research within the broader field.

**Questions:**

1. What is the cause of the sudden change behavior observed in the experimental results of the "origin" group in Figure 2?
2. Figure 2 serves as a clear and intuitive example. However, in more complex scenarios involving more than two possible safety areas, how is the behavior of the flipping-based policy?
3. The statement "Gaussian distribution-based stochastic policies are essentially deterministic policies disturbed by noise for exploration" may oversimplify the role and impact of Gaussian distribution-based policies. Is there some related work or more detailed explanations? While these policies (e.g. classical PPO and SAC) do indeed introduce stochasticity to facilitate exploration, they also fundamentally alter the optimization process during training. This change is not merely superficial but influences how the policy gradients are computed and how the agent learns to make decisions.

**Limitations:**

The authors thoroughly discussed the limitations and potential negative societal impacts in the Appendix.

---

> ### Author Rebuttal · Authors · 2024-08-05
>
> We appreciate the reviewer's valuable feedback and comments. We will answer the questions first and then address the comments about the weakness. All the concerns are addressed in a point-to-point way as follows.
>
> **Sudden change behavior [Question 1].**
> Thank you for the interesting question! The sudden change comes from the weakness of the original policy (deterministic policy). For a deterministic policy, if we set a small threshold of violation probability, passing the space between the two red-shaded circles will be infeasible since the violation probability will be larger than the threshold. Therefore, when increasing the threshold of violation probability from a very small value, the mean reward increases relatively slowly since the deterministic policies can only take the sideway in front of the two red-shaded circles. When the threshold of violation probability surpluses the lowest violation probability value of passing the middle space between the two red-shaded circles, passing the middle space will be feasible as this turning point, and the mean reward has a jump here. Note that the flipping-based policy can balance the violation probability by flipping between passing the middle space (high risk) and going the sideway (low risk). With the same mean risk, flipping can further improve the mean reward. **You can regard the flipping as a linear combination to establish a slope for the jumping**.
>
> **More complex scenarios [Question 2].**
> Thank you for this valuable comment. Due to our theoretical results (particularly Theorems 2 and 6), even in more complex scenarios involving more than two possible safety areas, **there exists an optimal solution, passing two of them randomly. These chosen "two" will be the optimal "two." The random choice will be made only from these chosen "two," depending on the flip probability.** In the PointGoal environment, there are far more than two possible safety areas. the flipping-based policy still improves the performance of CPO and PCPO. To validate the generality of adapting the flipping-based policy to the existing methods, we added the results of two new methods, CUP [Ref-1-1] and P3O [Ref-1-2]. Besides, for all four methods, we added the validations in another environment, CarGoal. The new results are summarized in a new One-Page PDF file attached to the global response. Please kindly check it.
>
> **Statement [Question 3].**
> Indeed, as the reviewer mentioned, the statement on Gaussian distribution-based stochastic policies is improper. Facilitating exploration is one of the benefits of introducing stochasticity. We want to emphasize the part of facilitating exploration and use the improper expressions that oversimplify the stochasticity's role. Thanks for pointing out this improper statement. We apologize for this improper statement and will correct it in the new version.
>
> **Typos [Weakness 1 and 2].**
> Thanks for pointing out the typos. We will correct them in the new version and check the whole manuscript to find and correct the other typos.
>
> **Assumption on reward and state transition function [Weakness 3].**
> Thank you for this insightful comment about the gap between the theoretical results and the general applications. The main contribution of this paper is theoretical. Currently, there is a gap in the scenarios with non-smooth functions. In future work, we will extend the results to more practical scenarios.
>
> **CCRL and ECRL [Weakness 4].**
> Thanks for your constructive advice on improving the coherence of the manuscript's presentation. When preparing the new version, we would like to follow your suggestion of focusing the discussion on the application of ECRL to CCRL. We will add one short paragraph in the introduction to emphasize this point and provide a short guide to help readers read the rest of the parts. To avoid misunderstanding, several changes will be made in the text, especially the part shifting focus to extending CCRL to ECRL.
>
> **Training process [Weakness 5].**
> As the reviewer mentioned, it is crucial to provide the training process. In a new One-Page PDF file attached to the global response, we have added the training process profile for each method. We will add these new results in the appendix in the new version.
>
> **Broader confidence intervals [Weakness 6].**
> Thank you for pointing out this critical point. From the experiment results in Omnisafe, the flipping-based policy introduces broader confidence intervals. For the numerical example in Section 5.1, the flipping-based policy does not give broader confidence intervals. The numerical example in Section 5.1 is closer to the optimal solution of the flipping-based policy. Thus, in theory, the flipping-based policy does not increase the size of the confidence interval. However, the practical implementation may have this issue. We think that it is possible to keep the confidence interval as the original one by using smaller variances in Gaussian noises, which may not hurt the performance of the mean reward and cost. The reviewer's observation is very important. We want to reflect this by a remark in the new version.
>
> **More environments [Weakness 7].**
> We added experimental validation results in a new One-Page PDF file attached to the "global" response. Except for the PointGoal environment, every method was validated in the CarGoal environment. The performance in the CarGoal environment is the same as that of the PointGoal environment.
>
> **Related work section [Weakness 8].**
> Indeed, having an independent part of related work can clearly explain how the current research fills the gap. We will extract parts of related work in the introduction and extend it into a related work section in the new version.
>
>
> [Ref-1-1]  Yang, L., et al, Constrained update projection approach to safe policy optimization. In NeurIPS, 2022.
>
> [Ref-1-2]  Zhang, L., et al, Penalized proximal policy optimization for safe reinforcement learning. In IJCAI, 2022.

---

### Author Rebuttal · Authors · 2024-08-05

Dear Reviewers and AC,

The authors deeply thank all the reviewers for their insightful comments and constructive suggestions.

1. We have conducted new experiments based on the reviewers' comments. Additional experimental results are provided in a One-Page PDF file containing new figures. The One-Page PDF file is attached in this global response;
2. We have provided our detailed response to each reviewer with a separate response.

We hope our responses have addressed all the concerns and questions of the reviewers.
We are willing to answer any of the reviewers' concerns about our work and sincerely wish the reviewers to value our paper's theoretical and technical contributions.

Best regards,

Authors

---

### Decision · Program_Chairs · 2024-09-25

**Decision:**

Accept (poster)

**Comment:**

This paper considers the problem of Chance-Constrained Markov Decision Processes (CCMDPs). The paper proposes a flipping-based policy that selects the next action by tossing a potentially distorted coin between two action candidates and the probability of the flip and the two action candidates vary depending on the state. The paper shows that a Bellman equation exists for CCMDPs, and using this, the paper shows the existence of a flipping-based policy within the optimal solution sets.  In terms of practical implementation, the paper proposes a framework for adapting constrained policy optimization to train a flipping-based policy. Using this framework, the proposed approach is demonstrated in Safety Gym benchmarks.

We received four expert reviews, with the scores, 5, 5, 7, 7, and the average score is 6.00. The reviewers are positive about the technical contribution and algorithmic novelty.

Reviewer g2rG and Reviewer GDWH have some questions about the simulation experiments, and I believe that the authors have addressed these questions satisfactorily in the rebuttal. Reviewer 7qnc and Reviewer 7QvG have commended the theoretical contributions. All the reviewers have some suggestions about improving the presentation of the paper. Please update the manuscript according to these comments while preparing the final version.